# Evaluating two soil carbon models within the global land surface model JSBACH using surface and spaceborne observations of atmospheric $CO_2$

Tea Thum[1], Julia E. M. S. Nabel[2], Aki Tsuruta[3], Tuula Aalto[3], Edward J. Dlugokencky[4], Jari Liski[3], Ingrid T. Luijkx[5], Tiina Markkanen[3], Julia Pongratz[2, 6], Yukio Yoshida[7], and Sönke Zaehle[1]

[1]Max Planck Institute for Biogeochemistry, Jena, Germany
[2]Max Planck Institute for Meteorology, Hamburg, Germany
[3]The Finnish Meteorological Institute, Helsinki, Finland
[4]NOAA Global Monitoring Laboratory, Boulder CO, United States
[5]Meteorology and Air Quality Group, Wageningen University and Research, Wageningen, The Netherlands
[6]Department of Geography, Ludwig-Maximilians-Universität, Munich, Germany
[7]Center for Global Environmental Research, National Institute for Environmental Studies, Tsukuba, Japan

**Correspondence:** Tea Thum (tthum@bgc-jena.mpg.de)

**Abstract.** The trajectories of soil carbon in our changing climate are of utmost importance, as soil is a substantial carbon reservoir with a large potential to impact the atmospheric carbon dioxide ($CO_2$) burden. Atmospheric $CO_2$ observations integrate all processes affecting carbon exchange between the surface and the atmosphere and therefore are suitable for carbon cycle model evaluation. In this study, we present a framework for how to use atmospheric $CO_2$ observations to evaluate two distinct soil carbon models (CBALANCE and YASSO) that are implemented in a global land surface model (JSBACH). We transported the biospheric carbon fluxes obtained by JSBACH using the atmospheric transport model TM5 to obtain atmospheric $CO_2$. We then compared these results with surface observations from Global Atmosphere Watch stations as well as with column $XCO_2$ retrievals from the GOSAT satellite. The seasonal cycles of atmospheric $CO_2$ estimated by the two different soil models differed. The estimates from the CBALANCE soil model were more in line with the surface observations at low latitudes (0°N-45°N) with only 1% bias in the seasonal cycle amplitude, whereas YASSO underestimated the seasonal cycle amplitude in this region by 32%. YASSO, on the other hand, gave more realistic seasonal cycle amplitudes of $CO_2$ at northern boreal sites (north of 45°N) with underestimation of 15% compared to 30% overestimation by CBALANCE. Generally, the estimates from CBALANCE were more successful in capturing the seasonal patterns and seasonal cycle amplitudes of atmospheric $CO_2$ even though it overestimated soil carbon stocks by 225% (compared to underestimation of 36% by YASSO) and its estimations of the global distribution of soil carbon stocks was unrealistic. The reasons for these differences in the results are related to the different environmental drivers and their functional dependencies of the two soil carbon models. In the tropics, heterotrophic respiration in the YASSO model increased earlier in the season since it is driven by precipitation instead of soil moisture, as in CBALANCE. In temperate and boreal regions, the role of temperature is more dominant. There, heterotophic respiration from the YASSO model had a larger seasonal amplitude, driven by air temperature, compared to CBALANCE, which is driven by

soil temperature. The results underline the importance of using sub-annual data in the development of soil carbon models when they are used in shorter than annual time scales.

## 1 Introduction

The terrestrial carbon cycle consists of uptake of $CO_2$ by vegetation for photosynthesis and release of carbon by plants' autotrophic respiration, soil decomposition by heterotrophic organisms and natural disturbances (Bond-Lamberty et al., 2016). Soils store twice as much carbon as the atmosphere (Scharlemann et al., 2014) and its fate in changing climate remains uncertain (Crowther et al., 2016). For example, while Crowther et al. (2016) concluded from a data-based analysis that large carbon stocks will lose more carbon due to warming conditions, van Gestel et al. (2018) questioned this view with an analysis based on a more comprehensive dataset. To have reliable predictions of future carbon stocks, process-based understanding of the below ground carbon cycle is needed (Bradford et al., 2016).

One way to evaluate soil carbon models has been to use observations of soil carbon stocks (Todd-Brown et al., 2013). At small scales, rates of gas exchange measured in chambers have also been used (Ťupek et al., 2019), but separation of heterotrophic and autotrophic respiration is laborious (Chemidlin Prévost-Bouré et al., 2010). It is anyhow challenging to find reasons for differences in heterotrophic respiration between large scale models, as the litter input to the soil influences heterotophic respiration and this litter input varies between the models. One way forward is to use a testbed for these models, as done by Wieder et al. (2018).

An alternative, regionally integrated approach is using observations of atmospheric $CO_2$, which integrate all processes involved in global surface-atmosphere carbon exchange. The surface observation network of atmospheric $CO_2$ has been used in benchmarking global carbon cycle models (Cadule et al., 2010; Dalmonech and Zaehle, 2013; Peng et al., 2015). Recent advances of satellite technology have enabled retrievals of space-born dry-air total column-averaged $CO_2$ mole fraction ($XCO_2$), quantifying $CO_2$ in the entire atmospheric column between the land surface and the top of the atmosphere. These observations reveal a more spatially integrated $CO_2$ signal compared to surface site observations and together they provide a complementary dataset. These two data sources have been used together to study the carbon cycle with "top-down" inversion modelling (Crowell et al., 2019). This kind of modelling framework uses atmospheric $CO_2$ observations to constrain a priori biospheric and ocean fluxes, based on the Bayesian inversion technique, which results in optimized estimates (a posteriori) of the fluxes (Maksyutov et al., 2013; Rödenbeck et al., 2003; van der Laan-Luijkx et al., 2017; Wang et al., 2019). Estimates for fossil emissions are often assumed as known, i.e., not optimized in the inversion.

In this study we present a framework of how to use atmospheric $CO_2$ observations to evaluate soil carbon models implemented in a land surface model. We apply this to two state-of-the-art soil carbon models as a "proof-of-concept" for a more universal application. Basile et al. (2020) did similar work within a biogeochemical testbed and concluded that heterotrophic

respiration can be a valuable benchmark in carbon cycle studies. They emphasized that the seasonal phasing of heterotrophic respiration relative to the net primary production influences the net ecosystem exchange and therefore potentially introduces bias to atmospheric $CO_2$ that hampers its use as a benchmark.

To obtain the atmospheric $CO_2$ profiles from our simulations with the land surface model we applied an atmospheric transport model. In this work we used a three-dimensional atmospheric chemistry transport model TM5 (Krol et al., 2005; Huijnen et al., 2010). Generally, transport models, such as TM5, contain errors caused by, for example, poorly resolved advection and heavily parameterized transport schemes (Gaubert et al., 2019). With TM5 we calculated the column averaged $CO_2$ that can be used to evaluate model results versus the satellite observations. Also satellite observations can include errors. The uncertainty for GOSAT observations has been estimated to be around 1 to 2 $ppm$ (Oshchepkov et al., 2013; Reuter et al., 2013). Contributors to uncertainties in the retrieval algorithms originate, for example, from the solar radiation database and handling of aerosol scattering (Yoshida et al., 2013). Last, also the column $XCO_2$ profiles have influences from, for example, advection and global scale gradients driven by weather systems (Keppel-Aleks et al., 2011). A model evaluation performed with the column $XCO_2$ observations enabled a more thorough study of fluxes and atmospheric physics of a modelling system (Keppel-Aleks et al., 2011).

We use in this work JSBACH, the land surface model of the Max Planck Institute's Earth System Model, one of the models participating in CMIP6. The JSBACH model has two distinct soil models implemented in it (CBALANCE and YASSO). We are interested in seeing if the two soil carbon models lead to markedly different $CO_2$ signals and to explore which conclusions on model performance and process representation can be drawn that could help to improve this land surface model (and potentially other similar models) and our understanding of the land carbon cycle. The two model versions only differ with respect to the underlying soil processes and do not include major feedbacks between soil and vegetation (apart from a small effect of litter accumulation on fire emissions). Thus, the difference in the release of carbon to the atmosphere originates only from the soil carbon models. The two soil carbon models are both first-order decay models. However, they have different pool structures as well al environmental drivers and have differing response functions. CBALANCE uses soil moisture and soil temperature as driving variables and YASSO precipitation and air temperature. In the analysis we also use a simple box model calculation to further understand the main causes in the different outcomes of the models. Our framework combining a land surface model with a transport model allows us to investigate how these above-mentioned differences in soil carbon models influence atmospheric $CO_2$. Specifically, we aim to answer the following questions:

- How can we use a land surface model together with a transport model to evaluate soil carbon models and what problems do we face when doing that?

- What is the role of soil carbon stocks, the variables driving their decomposition and the functional dependencies of those variables on modelled heterotrophic respiration at global scale and how does this lead to differences in the atmospheric $CO_2$ signal?

## 2 Materials and Methods

We used the land surface model JSBACH (Giorgetta et al., 2013) to obtain net land-atmosphere $CO_2$ exchange and fed that, together with ocean, fossil and land use fluxes, into a transport model, TM5, which simulates resulting atmospheric $CO_2$ at selected surface sites as well as column integrated values for comparison to satellite derived column $CO_2$.

### 2.1 Model simulations: JSBACH with two soil carbon models

JSBACH is the global land surface model of the Max Planck Institute's Earth System Model (Giorgetta et al., 2013), simulating terrestrial carbon, energy and water cycles (Reick et al., 2013). In this study JSBACH was run with two different soil carbon sub-models that are described below. The older model, CBALANCE, has been used in CMIP5 simulations of JSBACH (Giorgetta et al., 2013). The newer model, YASSO, has been used in simulations for the annual global carbon budget (Le Quéré et al., 2015; Le Quéré et al., 2018) and is used in CMIP6 simulations of JSBACH (Mauritsen et al., 2019). It is also used in JSBACH4, a re-implementation of JSBACH for the ICOsahedral Non-hydrostatic Earth system model (ICON-ESM) (Giorgetta et al., 2018; Nabel et al., 2019).

Independent of the sub-model used for soil carbon, JSBACH uses three carbon pools for living vegetation: a wood pool, containing woody parts of plants, and green and reserve pools that contain the non-woody parts. JSBACH simulates different processes that lead to losses from the vegetation pools, such as grazing, shedding of leaves and natural or anthropogenic disturbances. Depending on the process, some of the vegetation carbon is lost as $CO_2$ into the atmosphere, while the remaining part is transferred as dead vegetation into the litter and soil pools of the sub-model for soil carbon, where it is then subject to the internal processes of the soil carbon sub-model. The only process outside of the soil carbon sub-model that influences dead material is fire, burning parts of above ground litter carbon.

#### 2.1.1 The soil carbon model CBALANCE

CBALANCE (CBA) is the original soil carbon sub-model of JSBACH (Raddatz et al., 2007), which has been used in CMIP5. The environmental drivers for decomposition in CBA are soil temperature (at soil depth of 30 to 120 cm below the surface) and relative soil moisture ($\alpha$) of the upper-most soil layer, which is 5 cm thick. $\alpha$ varies between zero and one.

The function for soil temperature dependence, $f_{CBA,T_{soil}}$ of decomposition follows a $Q_{10}$ formulation as

$$f_{CBA,T_{soil}}(T_{soil}) = Q_{10}^{\frac{T_{soil}}{10^\circ C}} \tag{1}$$

with a $Q_{10}$ value of 1.8 and $T_{soil}$ as soil temperature in °C (shown in Fig. S1a) (Raddatz et al., 2007). The dependency on relative soil moisture $\alpha$ is linear (Fig. S1b) and it is calculated as

$$f_{CBA,\alpha}(\alpha) = MAX(\alpha_{min}, \frac{\alpha - \alpha_{crit}}{1.0 - \alpha_{crit}}) \tag{2}$$

where $\alpha_{crit}$ is 0.35 and $\alpha_{min}$ is 0.1 (Knorr, 2000).

Together these functions are modulating the rate of decomposition, so that the heterotrophic respiration ($R_h$) from each pool (denoted by $i$) is

$$R_h(T_{soil}, \alpha) = f_{CBA,\alpha} * f_{CBA,T_{soil}} * \frac{C_i}{\tau_i} \tag{3}$$

where $C_i$ is the carbon content of each pool and $\tau_i$ is the turnover time of each pool in days. CBA uses five different carbon pools having different turnover times:

- Two green litter pools: one above- and one below-ground in which the non-woody plant parts decompose with turnover times between 1.8 and 2.5 years (Goll et al., 2015)

- Two woody litter pools: one above- and one below-ground in which the woody plant parts are decomposed with turnover times of several decades

- One slow pool receiving its input from the four litter pools, having a turnover time in the order of a century.

### 2.1.2 The soil carbon model YASSO

The original soil carbon model of JSBACH was replaced by YASSO (YAS) (Thum et al., 2011; Goll et al., 2012). JSBACH's YAS implementation is based on the Yasso07 model (Tuomi et al., 2009). Development of Yasso07 relied heavily on litter bag and other observational data sets that were used to estimate model parameters (Tuomi et al., 2009, 2011). Owing to its strong connection to experiments, its environmental drivers are quasi-monthly air temperature and precipitation.

The decomposition dependency on air temperature is

$$f_{YAS,T_{air}}(T_{air}) = e^{\beta_1 T_{air} + \beta_2 T_{air}^2} \tag{4}$$

where $T_{air}$ is air temperature (°C), parameter $\beta_1$ is $9.5 \times 10^{-2}$ °$C^{-1}$ and parameter $\beta_2$ is $-14 \times 10^{-4}$ °$C^{-2}$ (Fig. S1c).

The decomposition depends on precipitation $P_a$ [m] as

$$f_{YAS,P_a}(P_a) = (1 - e^{\gamma P_a}). \tag{5}$$

where $\gamma$ = -1.21 $m^{-1}$ (Fig. S1d). The environmental drivers for YAS (precipitation and air temperature) are averaged for 30-day periods.

Similar to CBA, YAS has slowly and rapidly decomposing pools, but its pool dynamics are more structured. First, all the pools are divided into woody and non-woody materials. The difference in the calculation of the decomposition rates between non-woody and woody pools is an additional parameter that increases the turnover rates of the woody litter, dependent on its size parameter (Tuomi et al., 2011), which is plant functional type (PFT)-dependent.

YAS takes the chemical composition of the incoming litter into account. The incoming litter is divided to different chemical pools according to the PFT-dependent factors. Information for the PFT-dependent factors for the litter decomposition has been derived from observations (Berg, 1991; Berg et al., 1991; Gholz et al., 2000; Trofymow, 1998). YAS uses four chemically

distinct pools: acid soluble, water soluble, ethanol soluble and non-soluble. For each of these four chemical compositions one above- and one below-ground pool is used. In addition there is one humus pool (divided to woody and non-woody pools as all the other pools). Dynamics of the YAS carbon pools are described in Tuomi et al. (2009) with decomposition fluxes causing redistributions among the pools or losses to the atmosphere. Each of the pools has a decay constant, which is modified by the environmental dependencies in Eqs. (4) and (5).

## 2.2 The model simulations: The JSBACH set-up

JSBACH model simulations followed the TRENDY v4 protocol in terms of JSBACH version, simulation protocol and forcing data (Le Quéré et al., 2015; Sitch et al., 2015). Climate forcing was based on CRUNCEP v6 (Viovy, 2010) and global atmospheric $CO_2$ was obtained from ice core and National Oceanic and Atmospheric Administration (NOAA) measurements (Sitch et al., 2015). For each set-up, the model was run to equilibrium, i.e. until the soil carbon pools of the applied carbon sub-model were at steady-state. The two different transient simulations were then done for 1860 to 2014. Anthropogenic land cover change was forced by the LUHv1 dataset (Hurtt et al., 2011) and was simulated as described in Reick et al. (2013). While fire and windthrow were simulated, natural land cover changes and the nitrogen cycle were not activated. Simulations were done at T63 spatial resolution (approximately 1.9° or 200 km). For further details on the spin-up and the model version please refer to the SI.

## 2.3 The model simulations: TM5

To estimate atmospheric $CO_2$, we used the global Eulerian atmospheric transport model TM5 (Krol et al., 2005; Huijnen et al., 2010) in an available pre-existing set-up. TM5 was run globally at 6° x 4° (latitude x longitude) resolution with two-way zoom over Europe, where the European domain was run at 1° x 1° resolution. The 3-hourly meteorological fields from ECMWF ERA-Interim (Dee et al., 2011) were used as forcing to run TM5. Linear interpolation was done to obtain $CO_2$ estimates at the exact locations and times of the observations.

We fed TM5 daily biospheric as well as weekly ocean and annual fossil fuel fluxes to obtain realistic atmospheric $CO_2$. Values of gross primary production (GPP) and total ecosystem respiration were taken from the JSBACH simulations for the two different soil model formulations. Also, carbon release from vegetation and soil owing to land-use change, fires and herbivores were taken from the JSBACH model results as part of terrestrial biospheric carbon fluxes. In addition, we used the posterior biospheric flux estimates from CarbonTracker Europe (CTE2016, later referred to as CTE; van der Laan-Luijkx et al. (2017)) to provide some guidance on the ability of TM5 to represent the individual site observations. The ocean fluxes were the a posterior estimates from the same study.

Fossil fuel emissions are from the EDGAR4.2 Database (EDGAR4.2, 2011) and Carbones project (http://www.carbones.eu), with scaling to global total values as for the Global Carbon Budget as described in van der Laan-Luijkx et al. (2017). The annual fossil fuel flux to the atmosphere was approximately 8.63 $PgCyr^{-1}$, and ocean uptake of carbon was approximately 2.33 $PgCyr^{-1}$ when averaged over 2001-2014. Annual values are summarized in Table S1. Simulations with TM5 were done for 2000-2014.

## 2.4 The surface observations

Surface observations of atmospheric $CO_2$ from NOAA weekly discrete air samples (ObsPack product: GLOBALVIEWplus v2.1; ObsPack (2016)) were used to evaluate the effect of different soil carbon models on tropospheric $CO_2$ seasonal cycles at sites around the globe. The sites used in the evaluation are shown in Fig. 1. The uncertainties of NOAA flask air measurements for the period of this study are $\pm 0.07$ ppm (with 68% confidence interval). From the data, samples reflective of well-mixed background air were selected (based on flag criteria) similar to van der Laan-Luijkx et al. (2017) to minimize the influence of transport model errors in our analysis.

## 2.5 The satellite retrievals

GOSAT (Greenhouse Gases Observing Satellite) from Japan Aerospace Exploration Agency (JAXA) was launched in 2009 and observes column $XCO_2$ with the TANSO-FTS instrument (Kuze et al., 2009). These data were used to evaluate the different simulations and to assess model performance at larger spatial scale. $XCO_2$ from the TM5 simulation was calculated using global 4° x 6° x 25 (latitude x longitude x vertical levels) daily average 3-dimensional (3-D) atmospheric $CO_2$ fields. For each satellite retrieval, the global 3-D daily mean gridded atmospheric $CO_2$ estimates were horizontally interpolated to the location of the retrievals to create the vertical profile of simulated $CO_2$. Averaging kernels (AKs) (Rodgers and Connor, 2003) were applied to model estimates to ensure reliable comparison with GOSAT retrievals:

$$\hat{C} = c_a + (\boldsymbol{h} \circ \boldsymbol{a})^T (\boldsymbol{x} - \boldsymbol{x}_a), \tag{6}$$

where $\hat{C}$ is $XCO_2$, scalar $c_a$ is the prior $XCO_2$ of each retrieval, $\boldsymbol{h}$ is a vertical summation vector, $\boldsymbol{a}$ is an absorber-weighted AK of each retrieval, $\boldsymbol{x}$ is a model profile and $\boldsymbol{x}_a$ is the prior profile of the retrieval (Yoshida et al., 2013). The retrievals for different terrestrial TransCom (TC) areas (Fig. 1) were compared with those calculated from the two model simulations. For comparison with GOSAT $XCO_2$, the estimates of 3D fields at 6° x 4° resolution were used, but not those from the zoom grids due to technical reasons. Differences in $XCO_2$ due to model resolution were not significant within the context of this study. In this work GOSAT observations (NIES retrieval V02.21 and V02.31) between July 2009 and the end of 2014 were used.

## 2.6 Global datasets for evaluating simulated soil carbon and gross primary productivity

For evaluation of the JSBACH model results we additionally used data from two soil carbon databases and the FLUXCOM project (Jung et al., 2019). We used the gross primary production (GPP) produced by FLUXCOM, where eddy covariance flux observations are upscaled using machine learning methods and meteorological and remote sensing data. The FLUXCOM GPP has 0.5° spatial resolution and eight-day temporal resolution for 2001-2014. Additionally we used two different soil carbon datasets, SoilGrids (Hengl et al., 2014) and one based on Harmonized World Soil Database (HWSD) (Batjes, 2016). For the soil carbon data we used the preprocessed datasets from Fan et al. (2020) providing values for organic soil carbon down to 1 m depth.

## 3   Results

### 3.1   Global carbon fluxes and stocks with the two model formulations

#### 3.1.1   Carbon fluxes

Since the two different model formulations differ only in their soil carbon module formulation, the incoming flux to the ecosystem from photosynthesis is the same in both cases. We analyzed results for 2000-2014, and we show here averaged values for that period. The main target variable of our analysis is understanding the role of heterotrophic respiration, but to better elucidate how it influences the atmospheric $CO_2$, we also show net primary production (NPP) and net ecosystem exchange (NEE). NPP is obtained from the gross primary production (GPP) by subtracting autotrophic respiration. NEE is obtained by subtracting from GPP total ecosystem respiration, direct land cover change, fire, harvest and herbivory fluxes, as shown in Table 2.

Even though annual total global values of heterotrophic respiration are close between the different model formulations (Table 2), their global seasonal cycles are different (Fig. 2c). The YAS version has a 66% larger variation of $R_h$ during the year than CBA. Both model versions have their minimum value of $R_h$ in February. While CBA has a maximum in August, YAS reaches its maximum value one month earlier, and global $R_h$ also stays high during August. YAS clearly has a steeper increase and decline in its seasonal cycle than CBA. The higher peak of heterotrophic respiration by the YAS model leads to higher global NEE values during June and July (Fig. 2e). In the first four months of the year, NEE is higher in the simulations of the CBA model, caused by the higher heterotrophic respiration values at this time (Fig. 2e). Autotrophic respiration (which, as explained above, like GPP and NPP is the same for both model formulations) has its highest values in July and August (Fig. S2a). During 2000-2014 both CBA and YAS predict increases in heterotrophic respiration, but only YAS has a significantly increasing trend (p-value < 0.005) (Fig. 2). CBA has a larger standard deviation in the annual values (0.87 PgC) than YAS (0.73 PgC). The annual NEE time series do not have significant trends and CBA has larger interannual variability (standard deviation of 0.84 PgC vs. 0.79 PgC by YAS).

In addition to the comparison of the global results, we investigated how the two soil modules differed for broad latitudinally separated regions. The NPP is the same in the different latitudinal regions (Fig. 3a, b). The global total magnitudes of $R_h$ are comparable, while the seasonal cycles show clear differences, also visible in different latitudinal regions (Fig. 3c, d). The YAS model shows, however, a larger amplitude in the seasonal cycle in all of the regions. In the two most northern regions of the Northern Hemisphere the amplitude in $R_h$ of YAS is approximately twice the amplitude of CBA. In both of these regions YAS has clear maximum values of $R_h$ in July and August, while the seasonal cycles of CBA are more shallow and do not include such clear maximums. The seasonal cycle of $R_h$ is quite different between the model formulations in the tropics. At 0°N-30°N, YAS has a seasonal cycle shifted earlier compared to CBA. In this region YAS has a 42% larger seasonal amplitude for $R_h$ than CBA. In the Southern Hemisphere regions 0°S -10°S and 10°S -30°S, CBA predicts higher values of $R_h$ during the first months of the year after which it stays lower until the end of the year, whereas YAS shows a clear lowering between June

and September. In the region 10°S -30 °S YAS has 54% larger amplitude in $R_h$ than CBA. The differences in heterotrophic respiration lead to pronounced differences in the NEE within the tropics (Fig. 3e, f).

The variation in $R_h$ seasonal dynamics of these two model formulations can be linked to the differences in their environmental drivers and functions. In Table 3 the correlation between heterotrophic respiration and the environmental drivers of each specific model formulation are shown for the different latitudinal regions. Figures S3-S7 in the supplementary material show these same relationships. The $R_h$ from CBA has a strong correlation with soil moisture $\alpha$ in the tropical region (30°S-30°N) and a high correlation with soil temperature $T_{soil}$ in the northern high latitudes (30°N-90°N) and lower, but significant, correlation in southern high latitudes (30°S-60°S). For other combinations of regions and drivers the $r$ values are low for CBA and in three regions the dependency between $\alpha$ and $R_h$ is negative. In two of these regions with a negative relationship between $\alpha$ and $R_h$ (located in high latitudes), the variability of $\alpha$ is quite small and the plot shows high scatter (Fig. S3). The shape of the $T_{soil}$ dependency on the CBA decomposition is exponential, and the relationship is significant, when the range of the $T_{soil}$ values is over 15 degrees, which is larger than what is occurring in the tropics (Fig. S4).

For the YAS model, on the other hand, $R_h$ shows strong correlation to its environmental drivers (Table 3). The $r$ values between $R_h$ and precipitation are over 0.90 in all regions except region 30°S-60°S. In this region the correlation is still significant, but the variability of the precipitation is lower than in the other regions (Fig. S5). Therefore the exponential relationship (Fig. S1d, Eq. 5) between decomposition and precipitation does not lead to a stronger linear relationship in this region. Between air temperature and $R_h$ the results are similar, with the only small $r$ value in the Southern Hemisphere tropics. This region has only a small seasonal variation in air temperature and the values are also partly located in the temperature range, where the temperature sensitivity of decomposition is weaker (Fig. S6, Eq. 4). The seasonal cycle of $R_h$ predicted by the YAS model does not correlate significantly with the soil moisture variable $\alpha$ in any of these regions (Table 3 and Fig. S7). This is not unexpected as such, since $\alpha$ is not the driver of the YAS model. In the tropical region soil moisture for CBA and precipitation for YAS are more important drivers compared to soil and air temperatures. At high latitudes temperature has a larger effect on $R_h$ in the results of both models, even though in the Northern Hemisphere precipitation also has a significant role for YAS.

We also investigated, whether the seasonal cycle of the heterotrophic respiration is correlated with litter fall. The only significant correlation occurred at 30°N-60°N for both model versions. This was caused because both have similar annual cycles of $R_h$ and litter fall, but the seasonal cycle of $R_h$ precedes litter fall (Fig. S8).

Global simulated GPP of 167 $PgCyr^{-1}$ (Table 2) is highly overestimated when compared to the up-scaled data product from FLUXCOM, which is giving a mean value of 126 $PgCyr^{-1}$ for this time period (Jung et al., 2019) and having a range of 106-130 $PgCyr^{-1}$ for a longer time period. Despite the overestimation of global GPP by the model, the comparison to the FLUXCOM product shows that the seasonal cycles in different latitudinal regions are quite similar, although in the northern boreal region JSBACH reaches maximum GPP values later than the FLUXCOM product (Fig. S9).

The annual net $CO_2$ flux shows a slightly larger land sink for YAS than CBA (Table 2). Owing to the larger litter pool, fire fluxes are larger in the YAS model formulation by 0.50 $PgCyr^{-1}$, however they have similar spatial patterns (Fig. S10). This caused the heterotrophic respiration of YAS to be 0.56 $PgCyr^{-1}$ smaller than by CBA, since the model was spun-up to steady state in 1860 and thus leads to a small discrepancy in net $CO_2$ fluxes between the two model formulations.

### 3.1.2 Carbon stocks

The soil carbon stocks simulated by the two models differed in magnitude and also their latitudinal distributions differed. The global estimate for total soil carbon by CBA was 4.5-fold larger than by YAS (Table 1). The global estimate for litter simulated by the YAS model was larger than that simulated by CBA. Vegetation carbon biomass was similar in both model formulations (Table 1).

The global distribution of soil carbon is very different between the model formulations (Fig. S11c, d, Fig. S12). The CBA model has large values of soil carbon in the mid-latitudes of the Northern Hemisphere. YAS predicts larger values in the temperate region of the Northern Hemisphere, but the highest values of soil carbon are located in arctic regions. The data-based estimates from SoilGrids and HWSD also predict the highest values at high northern latitudes (Fig. S11a, b and Fig. S12). The CBA model predicts higher values and differing latitudinal pattern south of 60°N compared to the data-based values (Fig. S12). The YAS model shows very similar behaviour to the HWSD latitudinal pattern and magnitude south of 60°N. The $r^2$ and the root mean square errors are generally better for the YAS model than the CBA model when comparing the values along the latitudinal gradient against the data-based products (Table S2).

The turnover times of the two formulations must differ, since the soil carbon pools are of very different magnitude, but the annual $R_h$ between the model formulations are similar. The turnover times ($\tau$) of soil carbon pools can be evaluated at both grid scale and from global values. This global value is obtained by dividing the total soil carbon pool (to which both soil and litter carbon stocks are added) by the annual $R_h$. Calculated from the global values averaged for 15 years, the apparent turnover time for CBA is 51.3 years and for YAS 14.8 years. The turnover times of CBA are generally longer and show a large spread across different temperatures (Fig. 4 a). The YAS model shows a large spread of turnover times at warmer temperatures but below 0°C the range is narrower (Fig. 4 b). Both models predict the fastest turnover rates in moist and warm conditions. The anomalies of the turnover times are represented in Fig. 5. These have been calculated from the carbon pools over the whole time period and the mean annual $R_h$. The models show longer turnover times in northern high latitudes and dry areas. The CBA model shows longer turnover times in Central Asia, where the moisture conditions limit the decomposition. However, the YAS model doesn't show so large anomalies in this region.

### 3.1.3 Box model

To assess whether the larger seasonal cycle amplitude in $R_h$ by YAS is caused by the larger litter pool or the environmental response functions, a simple box model calculation was performed (detailed description is given in Appendix). When global respiration was calculated with the turnover times and soil carbon pools of the YAS model, but using the environmental responses and drivers of the CBA model, the annual magnitude decreased by 29% compared to the original YAS model (Table A1). However, the yearly maximum value did not change much. When the opposite was done, and the turnover time and soil carbon pools of CBA were used with the environmental responses and inputs of the YAS model, the magnitude of global heterotrophic respiration increased by approximately 1.4-fold (Fig. A1). The increase in the amplitude was 83% (Table A1). Therefore, this simple analysis suggests that the environmental variables and their response functions cause the larger

global amplitude of $R_h$ in the YAS model formulation. To further disentangle whether this change was caused by the different environmental drivers or their functional dependencies, we made additional tests.

The amplitudes of the seasonal cycle of $R_h$ (difference between the maximum and minimum values) are shown in Table A1. For the YAS model, there happens a strong decrease in the amplitude when both driver variables and the response functions are changed. When only driver variables are changed, only a slight decrease occurs. When the response functions are changed, the decrease in the amplitude is more pronounced with 21%. The amplitude predicted by the CBA model increases, when the driving variables and response functions are changed (Table A1). This increase occurs when either driving variables or response functions are changed individually. However, with the change of the response functions the change in amplitude is larger (74%). In summary, the response functions have a more pronounced role in the changes than the driving variables alone, and this was true for both models.

## 3.2    Evaluation against surface observations

Seasonal cycle amplitudes of atmospheric $CO_2$ are successfully simulated by the modeling framework across different latitudes (Fig. 6a). The $r^2$ values of the observed seasonal cycle and the model estimates are high across latitudes, despite some lower values in mid-latitudes of the Northern Hemisphere (Fig. 6b). Averaged over all latitudes the $r^2$ value, calculated as linear correlation of simulated and observed averaged annual cycles, was 0.93 for CTE, 0.90 for CBA and 0.87 for YAS.

The capability of the model formulations to simulate the amplitude of the seasonal cycle differs within latitudinal regions (Fig. 6). The CBA model is able to capture the timing of the seasonal cycle in northern latitudes, but has a tendency to overestimate the seasonal cycle amplitude by about 30% north of 45°N. In this region the underestimation of seasonal cycle amplitude by CTE is approximately 5% and by YAS 14%. In the region 0°N-45°N YAS underestimates the seasonal cycle amplitude, on average, by approximately 32%, whereas CTE underestimates it by 4% and CBA overestimates it by 1%. The agreement between estimated atmospheric $CO_2$ and observations was worse in YAS than in CBA when considering the $r^2$ value and the seasonal cycle. Overall, the magnitude of the seasonal cycle amplitude predicted by YAS had less bias north from 45°N compared to CBA, but large underestimation in latitudes 0°N-45°N, where CBA was very successful in simulating the right seasonal cycle amplitude.

Four surface observation sites in the Northern Hemisphere illustrate similar behaviour of the seasonal cycle and its amplitudes as described above (Fig. 7 and Table S3). To confirm the general quality of the TM5 model used for both YAS and CBA we plotted its biospheric posterior fluxes from CarbonTracker Europe 2016 (CTE); indeed, deviations between CTE and observations are much smaller than from the JSBACH model at all sites. At the high-latitude sites, Alert and Pallas (Fig. 7a, e), CBA overestimates the seasonal cycle amplitude, while YAS shows some phase-shift of the cycle. The observed seasonal cycle amplitudes are smaller at the two more southern sites, Niwot Ridge and Mauna Loa. For those sites, CBA is generally successful in capturing their magnitude (Table S3), whereas YAS underestimates them strongly. YAS is also having difficulty capturing the seasonal pattern at Niwot Ridge. This was happening generally in the temperate region, as is also seen in the lower $r^2$ values of the YAS model at the different sites (Fig. 6).

When comparing the overall bias in atmospheric $CO_2$ at these four sites between the observations and the model simulations, CBA overestimated $CO_2$ by 3.65 ppm and YAS by 2.27 ppm, when averaged over all the measurements within the study period. A closer look at the bias at Mauna Loa (Fig. S13) revealed biases in the 2000-2014 trends for CBA and YAS, whereas CTE shows no bias in trend. The CBA overestimates $CO_2$ by 1.76 ppm in the beginning and by 3.74 ppm in 2014. The overestimates by YAS are smaller, 1.12 ppm in 2000 and 3.14 in 2014. The results at surface sites show that CBA largely overestimated seasonal cycle amplitude at high northern latitudes, whereas YAS almost consistently underestimated the seasonal cycle amplitude in the Northern Hemisphere. CBA captured the seasonal cycle patterns better than YAS across different latitudes. Overall, the YAS model showed biases in the atmospheric $CO_2$ cycle at temperate latitudes in the Northern Hemisphere, whereas the CBA model had biases in the high latitudes in the Northern Hemisphere.

### 3.3 Column $XCO_2$ comparisons for TransCom regions

This evaluation of the two soil modules against satellite column $XCO_2$ was carried out for the different TransCom (TC) regions (Fig. 1). The comparison was based on seasonal cycle amplitudes and $r^2$ values similar to the surface site evaluation. Not all the TC regions show a clear seasonal cycle, such as regions in South America (TC regions 3 and 4), northern part of Africa (TC=5) and Australia (TC=10). For completeness we show the analysis also for these regions in Table S5. For regions with clear seasonal cycles we used the ccgcrv curve fitting procedure available from NOAA (https://www.esrl.noaa.gov/gmd/ccgg/mbl/crvfit/crvfit.html, Thoning et al. (1989)), but for regions with missing data or no clear seasonal cycle, we averaged over all years of data.

To further illustrate the results from this comparison, we show data for two regions having a clear seasonal cycle. In TC region 2, the southern part of North America, CBA is more successful in capturing the observed seasonal cycle amplitude than YAS (Fig. 8a), even though CBA reaches the minimum $XCO_2$ later than observations. YAS underestimates the seasonal cycle amplitude by 56% and has a different seasonal pattern than observations, so the minimum is reached earlier than in the observations and also the shape during the summer period differs from the observations. In Europe, TC region 11, both models capture the seasonal cycle amplitude (Fig. 8b, Table S4) and the seasonal cycle in the first part of the year. The increase of $CO_2$ in autumn is not captured so well by the simulations.

Overall, observed and simulated $XCO_2$ differ from each other in ways similar to the surface site observations. Estimates of seasonal cycle amplitude by YAS are too small in mid-latitudes (Fig. 8a) and in TCs 2, 5 and 8 compared to the observations, and CBA is better at capturing the observed annual cycles. At TC=1 (the northern part of North America), CBA overestimates the seasonal cycle amplitude, while YAS better captures it. However, the seasonal cycle pattern is better captured with CBA (Table S4) than with YAS. Generally YAS had smaller seasonal cycle amplitudes than the observations and CBA was more consistent with the observations in most TC regions. CBA is also better than YAS in capturing the seasonal pattern of $XCO_2$ in all TC regions (Table S4).

There is bias in absolute $XCO_2$ between the GOSAT retrievals and the model simulations. When averaged over the time period used and the TC regions, CBA overestimates the GOSAT observations by 3.37 ppm and YAS by 2.33 ppm. These values were in line with bias in absolute $CO_2$ estimates at the four surface sites.

## 4   Discussion

In this work our aim was to use atmospheric observations to assess whether soil carbon models of a land surface model can be evaluated with this kind of framework. Our main finding was that the two models predicted different annual cycles of global $R_h$, with the YAS model having a larger amplitude. This in turn leads to clear differences in the model predictions of seasonal cycles of the atmospheric $CO_2$ concentration at surface stations and TC regions. To attribute the differences between the two models to a specific cause, we need to compare their results from their different aspects and to also judge whether our model simulations are reasonable in the light of previous research.

### 4.1   Evaluation of carbon fluxes

Annual heterotrophic respiration was 66.1 $PgCyr^{-1}$ for CBA and 65.5 $PgCyr^{-1}$ for YAS (Table 2), which falls in the range of estimates from Earth System Models (41.3-71.6 $PgCyr^{-1}$) and is close to the observation based estimates of 60 $PgCyr^{-1}$ (Shao et al., 2013). Part of the difference between CBA and YAS is caused by the fire fluxes. The YAS model has a larger litter pool that behaves as fuel for fires. Therefore, to have the system at steady state, global heterotrophic respiration by YAS must be less. Moreover, the simulation time of 140 years before the beginning of the analysis might cause some divergence between the model runs.

Moving to monthly time scales, we can see that the global seasonal $R_h$ cycle had a larger amplitude with YAS than with CBA (Fig. 2) and a simple box model calculation found that environmental drivers and their response functions are the cause, not the large litter pool in the YAS model. It is anyhow challenging to further disentangle whether this larger amplitude is mainly caused by the differing environmental drivers of the soil carbon models or if the functional dependencies of those drivers would play a bigger role. The analysis by the box model suggested a stronger role of the response functions compared to the driving variables at monthly timescales, but strong conclusions cannot be drawn from such a simple analysis. Also other studies have showed that the response functions themselves lead to pronounced differences between soil carbon models (Wieder et al., 2018).

When heterotrophic respiration is compared by latitudinal zones, differences between the model formulations are visible in the variability and timing of the seasonal cycles in many regions (Fig. 3). $R_h$ correlates strongly with the environmental drivers of the models in different latitudinal zones (Table 3). Both models are largely influenced by their moisture dependency in the tropical region (Table 3). CBA is driven by soil moisture with a linear dependence and YAS is driven by precipitation with an exponential relationship. Since the ranges of precipitation are larger than the variability in soil moisture and due to the exponential relationship between precipitation and decomposition in YAS, YAS is more tightly coupled to moisture than CBA. At annual timescales, at which the YAS model was originally developed, precipitation and soil moisture behave similarly. However, the seasonal cycles of the two variables are different. Precipitation begins earlier in the season in the tropical region, and it causes YAS to reach yearly maximum heterotrophic respiration earlier than CBA, which is driven by soil moisture in this region. Similarly, air and soil temperatures are more similar on the long term as for short periods. Particularly in the temperate

region, where the temperature has a larger role, the air temperature has larger variability than soil temperature and this leads to different kind of seasonal pattern of the $R_h$ predictions by the two different soil models.

The observations show that litterfall has strong influences on heterotrophic respiration (Chemidlin Prévost-Bouré et al., 2010). At seasonal timescales in the different latitudinal zones, there is no clear influence of litterfall driving the heterotrophic respiration seen in the models, which primarily results from the pre-defined turnover times of the fast litter pools, which smooth out individual litter fall events. Changes in the chemical composition of litterfall are considered to be one potential reason for changes in the amplitude of atmospheric $CO_2$ (Randerson et al., 1997) and this is something we could study with the YAS
model.

Different moisture dependencies of $R_h$ have earlier been found to be important (Exbrayat et al., 2013). At the global level Hursh et al. (2017) recommended using parabolic soil moisture functions in preference to functions based on mean annual precipitation. Their study considered soil respiration, i.e., autotrophic respiration by roots was also included. Ťupek et al. (2019) evaluated the YAS model against $R_h$ observations at two coniferous sites in southern Finland and found problems in capturing
the seasonality in the observations and the variability in the summertime fluxes. One reason they mention for this is response of the simulated $R_h$ to soil moisture conditions, since $R_h$ is not attenuated in very moist conditions and they found a need to improve the moisture dependency of the YAS model. This is in line with our findings, that a model that has been parameterized at annual time scales requires further development before it can be reliably applied at shorter timescales. Precipitation was originally used in the YAS model as a proxy for soil moisture, since enough accurate soil moisture observations for model
development were not available. Clearly, this idea needs reconsideration as our results show that at zonal spatial scales and monthly temporal scale, $R_h$ from YAS is not correlated to the soil moisture.

Simulated global GPP (165 $PgCyr^{-1}$) is notably larger than the estimated 106-130 $PgCyr^{-1}$ derived from FLUXCOM for the time period. However, the simulated value is still within the range of other data-driven estimates such as the one from Carbon Cycle Data Assimilation system, being 146 ($\pm$ 19) $PgCyr^{-1}$ (for 1980-1999) (Koffi et al., 2012) and isotope based
estimates of 150 to 175 $PgCyr^{-1}$ (for 1980-2009) (Welp et al., 2011). Fig. S9 shows that the bias relative to FLUXCOM exists throughout most of the Northern Hemisphere and the tropics, but has only minor influence on the seasonal cycle of GPP. The high estimate of GPP will propagate into larger NPP, litter input and therefore also simulated heterotrophic respiration and soil carbon stocks. While this may contribute to a slightly larger simulated seasonal cycle of atmospheric $CO_2$ at northern stations, it is unlikely that this will affect our conclusions on the impact of the different soil formulations on the ability of JSBACH
to simulate the seasonal cycle of heterotrophic respiration and the residence time of carbon in soil, and as a consequence, its ability to reproduce observed seasonal cycle of atmospheric $CO_2$ or its longterm trend. Nevertheless, this comparison shows that in order to further improve JSBACH's performance against these data, GPP biases should be reduced. Furthermore, the high GPP values resulting from the simulations would likely be lower, if the nutrient cycles of nitrogen and phosphorus were included in the used version of JSBACH (Goll et al., 2012). Beside using a JSBACH version with nutrient cycles, further
development work in the phenological cycle could improve the estimated GPP. The difference of the modelled GPP to the FLUXCOM product (Fig. S9) suggests that the maximum leaf area index might be overestimated in the tropics. Also, the timing of the phenological cycle north of 60°N might benefit from re-parametrization.

## 4.2 Evaluation of carbon stocks and turnover times

The two soil models predicted different global soil carbon stocks (Table 1) with different latitudinal distributions (Fig. S12). Similar to earlier studies (Goll et al., 2015; Thum et al., 2011), in our results the YAS model was more successful than CBA in estimating global soil carbon stocks similar to estimates from observations, approximately 1500 PgC including large uncertainties (from 504 to 3000 PgC) (Scharlemann et al., 2014), as can be seen in the different estimates from HSWD (1578 PgC) and SoilGrids (2870 PgC) (see also Tifafi et al. (2018)). The YAS model is widely used in different applications at smaller scale and its performance to estimate soil carbon stocks has been found to be good (Hernández et al., 2017). Comparability between the model-calculated and the observed carbon stocks is relevant for any analyses of carbon fluxes because in both models investigated here the fluxes are proportional to the stocks (flux = decomposition rate * stock). Modelled global vegetation carbon was within the observation-based estimate of $442 \pm 146$ PgC by Carvalhais et al. (2014).

The distribution of soil carbon stocks was also more realistic in YAS than in CBA (Fig. S12, Table S2). The large soil carbon stocks in the mid-latitudes predicted by CBA (Figs. S11c, S12) are unrealistic compared to current data-based estimates of the global soil carbon distribution (Fig. S12). The large carbon stocks at high latitudes predicted by the YAS model (Figs. S11d, S12) are more in line with the observations, but miss the high values observed from peatlands and permafrost in high latitude regions. The version of JSBACH used does not include peatlands and is modelling only mineral soils. Therefore, the large carbon reservoirs of peatlands are not captured by the model. This JSBACH version also didn't have permafrost described. If permafrost would be modelled, the seasonal cycle of heterotrophic respiration at high latitudes would likely be dampened, as the depth of the active layer determines the amount of soil capable of respiring. The YAS model has been used in a JSBACH version containing permafrost in a study concentrating on the Russian Far East (Castro-Morales et al., 2018). Both, CBA and YAS, were originally developed for mineral soils and for applications with organic soil, so model development and testing at smaller than global scale could be useful.

The environmental responses of the turnover times have quite different forms for the two soil carbon models (Fig. 4). The CBA model shows a wide distribution of turnover times across the whole temperature range, whereas the YAS model shows a larger spread in the tropical temperature range. This large spread in warm conditions is also observed (Koven et al., 2017) and is caused by the saturating temperature function of the YAS model, as shown in Fig. S1c. The large spread in turnover times as predicted by the CBA model might be caused by the fact that CBA is driven by soil temperature in one soil layer. The environmental responses of the turnover times at annual time scales behave similarly as at monthly time scales, so that moisture is a more important driver in warm regions and temperature in cold regions, as was seen in Table 3.

The study by Koven et al. (2017) provided an empirically based turnover time as a function of temperature. At 20 °C this turnover time was approximately $11 \pm 2$ years, being closer to the estimate for the YAS model (calculated for values 19.5 - 20.5 °C, and their standard deviation), being $22 \pm 21$ years °C and much lower compared to the CBA estimate of $64 \pm 37$ years. In lower temperatures, at -15 °C, the empirically based turnover time is $200 \pm 100$ years, and YAS underestimates this with $82 \pm 41$ years (calculated for values -15.5 - (-14.5) °C), whereas the prediction by CBA is closer ($150 \pm 80$ years). Therefore, the turnover times simulated with the YAS model are closer to the observations in warm temperatures, but the turnover times

are too low in cold temperatures. CBA estimated too high turnover times in warm temperatures, but turnover times in colder temperatures were in the same order as the observations.

The global turnover time of soil carbon by CBA was somewhat larger than in an earlier study, where it was estimated to be 40.8 years (Todd-Brown et al., 2014). This value was in the higher end of the CMIP5 models. The global turnover time from YAS, which was 14.8 years, is more in the range of the other CMIP5 models (Todd-Brown et al., 2014). The spatial distribution of the turnover time anomalies show differences caused by the environmental drivers and their dependencies at annual timescales. When comparing these overall turnover times of total soil carbon, it is important to keep in mind that both models consisted of carbon pools that had widely varying turnover times. For example, despite the higher overall turnover time, the turnover time of the most recalcitrant carbon pool of YAS was an order of magnitude smaller than that of CBA.

## 4.3 Evaluation using atmospheric $CO_2$

The differences between the two models in the seasonal cycle of atmospheric $CO_2$ were strong. CBA better reproduced the seasonal cycle amplitudes capturing the shape of the seasonal cycle both for surface sites and comparisons in the TC regions, even though its soil carbon distribution had worse performance compared to YAS. CBA exaggerated the seasonal cycle amplitudes at high northern latitudes, as has been found earlier (Dalmonech and Zaehle, 2013). It is important to keep in mind that this study was done within a land surface model and modelled GPP was biased. The simulated GPP had a larger magnitude and some bias in its seasonal cycle, and therefore its evaluation against atmospheric $CO_2$ observations is influenced by it. Even though the atmospheric observations provide a valuable and informative comparison for the model results, their use as a benchmark metric needs careful consideration.

The differences in absolute $CO_2$ and $XCO_2$ levels against the surface observations and the satellite retrievals, respectively, with modelled $CO_2$ are caused by the modelling system, but this bias does not influence the analysis performed. We obtained the land surface fluxes (GPP, respiration, fire, herbivory fluxes, land-use change emissions) from JSBACH and together with the rest of fluxes from CarbonTracker Europe2016 (CTE), we used TM5 to obtain atmospheric $CO_2$ values. Fossil fuel emissions have not been optimized in CTE. Therefore we obtained ocean fluxes that had been optimized with the land carbon cycle of CTE, that differ from the JSBACH estimate. The land carbon cycle of CTE is modelled by the SiBCASA-GFED4 model (van der Velde et al., 2014) and fire emission that were estimated from satellite observed burned area (Giglio et al., 2013). The net global a posteriori land sink of CTE is approximately -2.0 ($\pm$ 1.1) $PgCyr^{-1}$ for 2001-2014. On the other hand, the JSBACH estimate for the net land sink is approximately -1.7 $PgCyr^{-1}$ (Table 2) and is therefore smaller than the land sink by CTE. The fire flux of JSBACH is modelled, whereas the estimate in CTE is based on data. As shown in Fig. S13 for Mauna Loa, the bias in the $CO_2$ develops during the study period and the plot shows consistency so that YAS, which predicts a net land sink closer to CTE than CBA, has smaller bias at the end of the time period. We concentrated the analysis on the averaged seasonal cycles, that are not influenced by this linear increase.

The space-borne observations give a similar message as the surface observations in TransCom regions, which showed clear seasonal cycle. Niwot Ridge is located in TransCom region 2 (southern part of North America) and also there YAS showed too low amplitude and CBA performed better, similarly as seen in the Fig. 8. The Pallas site is located in TransCom region

11 (Europe) and at Pallas the seasonal cycle was more pronounced than in Europe as whole, but similarly for the surface observations at Pallas and TransCom region 11, the models both perform acceptably. Using large TransCom regions helped to interpret the signal despite the larger variability than in the surface observations (comparing grey shaded regions in Figs. 7 and 8) and it has been recommended to use the information content of the satellites on continental scales (Miller et al., 2018).

The transport model itself also brings uncertainty to the result. Modelling of atmospheric transport is a challenging task as open scientific questions in the field remain (Crotwell and Steinbacher, 2018) and the models contain biases (Gurney et al., 2004). The errors in atmospheric transport models cause a substantial difference in the inverse $CO_2$ model flux estimates (Peylin et al., 2013). However, in this study we only used one atmospheric transport model. It is expected that the biases, as only one transport model was used, are similar between the two soil model runs and are not the cause for the large differences
seen in the two simulations.

## 5    Conclusions

We demonstrated how atmospheric $CO_2$ observations can be used to evaluate two soil carbon models within the same land surface model and the different viewpoints offered by several variables considered. We used two different soil carbon models within one land surface model and used a three-dimensional transport model to obtain atmospheric $CO_2$, while obtaining the
anthropogenic and ocean fluxes from CarbonTracker Europe framework. We evaluated the carbon stocks of the soil models and compared seasonal cycles calculated with soil carbon fluxes from the soil models to atmospheric $CO_2$ results from both surface and space-born observations. This work highlighted how the changes in the heterotrophic respiration transfer to the net ecosystem exchange estimates and further to the atmospheric $CO_2$ signal. We also discussed the importance of the model drivers and their functional dependencies, which differed for the two soil carbon models we studied. When considering both
surface- and space-based observations, it is not straightforward to say which of the two soil carbon models performed better.

The comparison of the two soil carbon models revealed large differences in their estimates. The YAS model better captured the magnitude and spatial distribution of soil carbon stocks globally. However, it was biased in its atmospheric $CO_2$ cycle at temperate latitudes in the Northern Hemisphere. The CBA model, on the other hand, showed better performance in capturing the seasonal cycle pattern of atmospheric $CO_2$, but it is biased at high latitudes in the Northern Hemisphere. $R_h$ from the
YAS model showed misalignment with soil water content in tropical regions, as they were negatively correlated with each other. This suggests that use of precipitation as a proxy for soil moisture might not be sensible at sub-annual time scales and calls for improvement in the parameterization of the YAS model. The use of this modelling system can help to assess the global consequences of the new YAS parameterization, if such is made. The drivers of YAS have larger variability in their values during the seasonal cycle, that causes a more pronounced seasonal cycle in the heterotrophic respiration with the current
parameterization. Concerning the results this leads to unrealistic seasonal cycles of $CO_2$ in temperate regions and tropics and calls for model improvement. CBA showed less pronounced seasonal cycles of heterotrophic respiration, and had issues with $CO_2$ amplitude only in the northern high latitudes. The linear moisture dependence therefore seems justified, however it likely causes the Central Asian region to have too large carbon stocks. Whether this is caused by too high drought sensitivity or

problems in the predicted soil moisture by JSBACH is difficult to judge. The too high amplitude in the northern high regions
might be a result from the biases in the gross fluxes of the modeling system.

The evaluation was done within a land surface model that overestimates GPP in comparison to an upscaled GPP product and this hampers doing benchmarking using this modeling system. Since the model is run to a steady-state during the spin-up procedure, it also leads to other biases in the modelling system (influencing e.g. autotrophic respiration). Overestimated GPP leads to an enhanced litter input to the soil. This causes comparing the magnitudes of the soil carbon pools to the actual observations cumbersome, as the overestimated litter fall causes biases in the model estimates. In this study the magnitudes of simulated soil carbon are therefore not as good as the spatial patterns as an indicator for the model performance (such as latitudinal gradient). The other downside of the GPP biases is their influence on the estimated NEE. Due to the biases in the timing and magnitude of the other carbon fluxes, it is challenging to use $CO_2$ as a benchmark to heterotrophic respiration. However, in our study the two soil models lead to pronounced differences in the atmospheric $CO_2$ and we were also able to locate latitudinal regions, where the models had most issues. Therefore, this approach provides a method to evaluate how the changes in the heterotrophic fluxes further influence the atmospheric signal and helps to track which geographical areas are contributing to the questionable model performance.

Soil carbon models have several development needs (Bradford et al., 2016; van Groenigen et al., 2017) that are now partly being answered with next generation models including more mechanistic representation of several below ground processes (Wieder et al., 2015; Yu et al., 2019). The development of moisture dependency from simple empirical relationships is moving towards mechanistic approaches, which may yield more reliable results in the long term (Yan et al., 2018). Our results confirm that the moisture dependency of heterotrophic respiration plays on important role in the whole global carbon cycle.

In this study we used space-born $XCO_2$ observations in addition to the surface observations of $CO_2$. They were providing a larger-scale confirmation for the results obtained from the surface observations and thus provided complimentary information. The number of satellite observations of column $XCO_2$ are increasing at a fast pace for example OCO-2 observations started in 2014, and they possess high potential for carbon cycle studies (Miller and Michalak, 2020).

*Code and data availability.* The site level data from Global Atmospheric Watch -network is available via Obspack (2016) (https://doi.org/10.15138/G3059Z). The EDGAR4.2 emission database is available at http://edgar.jrc.ec.europa.eu. The GOSAT data are from GOSAT Data Archive Service (GDAS) (https://data2.gosat.nies.go.jp/index_en.html). The CRUNCEP data is available from Viovy (2010) (https://vesg.ipsl.upmc.fr/thredds/catalog/store/p529viov/cruncep/V7_1901_2015/catalog.html). The JSBACH model can be obtained from the Max Planck Institute for Meteorology, and it is available for the scientific community under the MPI-M Sofware License Agreement (http://www.mpimet.mpg.de/en/science/models/license/, last access: 16 September 2019). The CarbonTracker Europe code is continuously updated and available through a GIT repository at Wageningen University and Research: https://git.wur.nl/ctdas. For further details, see also: www.carbontracker.eu. The transport model TM5 is available via https://svn.knmi.nl/svn/TM5. For the curve fitting for the atmospheric $CO_2$ data we used scripts available from ERSL NOAA at https://www.esrl.noaa.gov/gmd/ccgg/mbl/crvfit/crvfit.html.

## Appendix A: Description of the box model

A simple box model calculation was performed to evaluate the importance of the dependencies of environmental drivers and the soil carbon pool sizes on the larger global seasonal cycle amplitude in $R_h$ as predicted by YAS. In this box model, we assume that heterotrophic respiration $R_h$ is a product of environmental dependencies and the turnover time as

$$R_{h,YAS} = b * f_{YAS,T_{air}}(T_{air}) * f_{YAS,P_a}(P_a) * \frac{C_{soil,YAS}}{\tau_{YAS}}, \text{where} \qquad b = \frac{\Sigma f_{CBA,T_{soil}}(T_{soil}) * f_{CBA,\alpha}(\alpha)}{\Sigma f_{YAS,T_{air}}(T_{air}) * f_{YAS,P_a}(P_a)}, \tag{A1}$$

where $R_{h,YAS}$ is the heterotrophic respiration of model YAS, $b$ is a scalar that takes into account the different magnitudes of the response functions, $T_{air}$ is air temperature, $P_a$ is annual precipitation, $C_{soil,YAS}$ are the total soil carbon pools and $\tau_{YAS}$ is the turnover time of the total soil carbon pools. $T_{soil}$ is soil temperature and $\alpha$ is the relative soil moisture. This formulation in A1 refers to the YAS model. The response functions are as shown in Section 2.1.2. For the CBA model the formulation is as

$$R_{h,CBA} = \frac{1}{b} * f_{CBA,T_{soil}}(T_{soil}) * f_{CBA,\alpha}(\alpha) * \frac{C_{soil,CBA}}{\tau_{CBA}}. \tag{A2}$$

These responses were introduced in Section 2.1.1.

The equations used monthly heterotrophic respiration, environmental drivers and soil carbon stocks averaged over 2001-2014 to estimate the turnover times for each grid point for YAS using Eq. A1 and for CBA using Eq. A2. Using these turnover times, we calculated global $R_h$ with the turnover times and soil carbon pools of each model by making different tests. First, we used the environmental responses and drivers of the other model (lines B in Table A1). Additionally we changed the driving variables, but kept the original response functions (lines C in Table A1). Then we changed only the response functions of the original model while keeping the original driving variables (lines D in Table A1).

Since the driving variables of soil moisture and annual precipitation differed in magnitude by approximately 4-fold, soil moisture was multiplied by four when using the function for annual precipitation ($f_{YAS,P_a}$) and when annual precipitation was used in the function for soil moisture ($f_{CBA,\alpha}$) it was divided by four. The annual cycles of $R_h$ are shown in Fig. A1 and the amplitudes in Table A1.

*Author contributions.* TT designed the experiment with the help of SZ. JEMSN performed the JSBACH model simulations. AT did the CarbonTracker Europe (CTE2016) runs with the JSBACH biospheric fluxes, with the $CO_2$ fields provided by ITK. ITK provided the CarbonTracker Europe (CTE2016) results used for comparison at the surface stations. TT performed the analysis with help from SZ, AT and TM. TT wrote the first version of the draft and all the authors contributed to the manuscript.

*Competing interests.* Dr. Sönke Zaehle is an associate editor for Biogeosciences.

*Disclaimer.* TEXT

*Acknowledgements.* TT was funded by Academy of Finland (grant no. 266803). TT and SZ were funded by European Research Council (ERC) under the European Union's Horizon 2020 research and innovation programme (QUINCY; grant no. 647204). SZ was furthermore supported by the European Union's Horizon 2020 Project funded under the programme SC5-01-2014 (CRESCENDO, grant No. 641816). ITL received funding from Netherlands Organisation for Scientific Research (NWO) under contract no. 016.Veni.171.095. JEMSN and JP were supported by the German Research Foundation's Emmy Noether Program (PO1751/1-1). JSBACH simulations were conducted at the German Climate Computing Center (DKRZ; allocation bm0891). We thank Dr. Janne Hakkarainen for helping in analysing the GOSAT data and averaging kernel calculation. We thank Dr. Martin Jung for access to the FLUXCOM results and the FLUXCOM initiative. We are grateful for Naixin Fan for sharing the preprocessed SoilGrids and WHSD data with us. We thank Prof. Dr. Wouter Peters for constructive comments on an earlier version of this manuscript. We thank Dr. Willy R. Wieder and one anonymous reviewer whose constructive comments improved this manuscript.

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

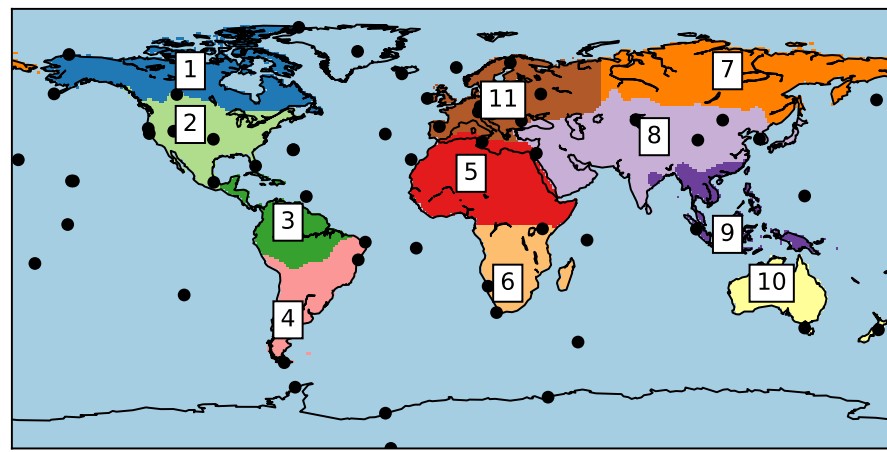

**Figure 1.** Locations of Global Atmosphere Watch stations, denoted as black dots, and different TransCom regions (different numbers denote the different TransCom regions in this study) as different colors.

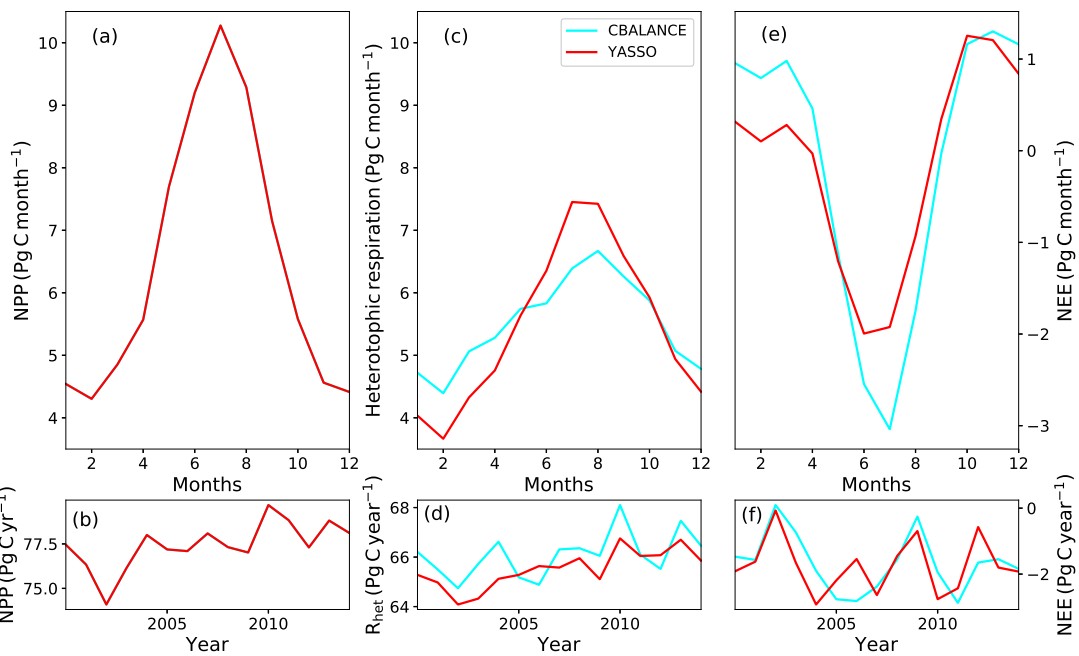

**Figure 2.** The annual cycles of net primary production (NPP) (a), heterotrophic respiration (c) and net ecosystem exchange (NEE) (e) globally with the CBALANCE (in cyan) and YASSO (in red) model versions. In the subpanels, annual values are plotted for 2000-2014 for NPP (b), heterotophic respiration (d) and NEE (f).

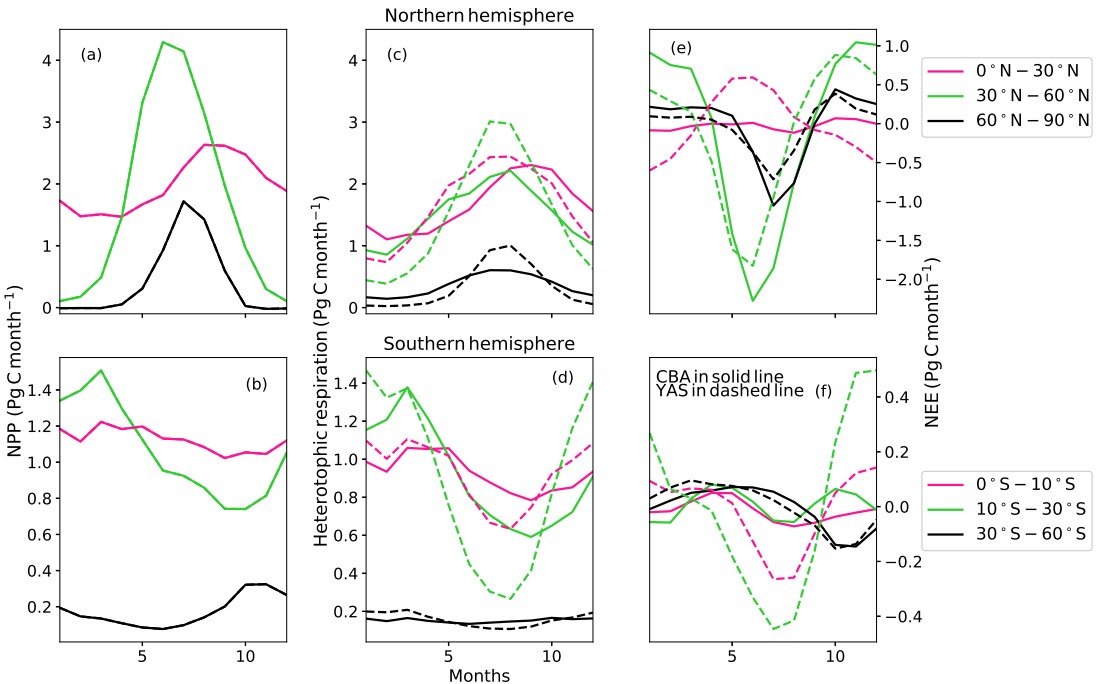

**Figure 3.** The annual cycle of net primary production (NPP) (a, b), heterotrophic respiration (c, d) and net ecosystem exchange (e, f) in Northern and Southern Hemispheres separated into latitudinal zones. CBALANCE (CBA) results are shown in solid lines and the YASSO (YAS) results in dashed lines.

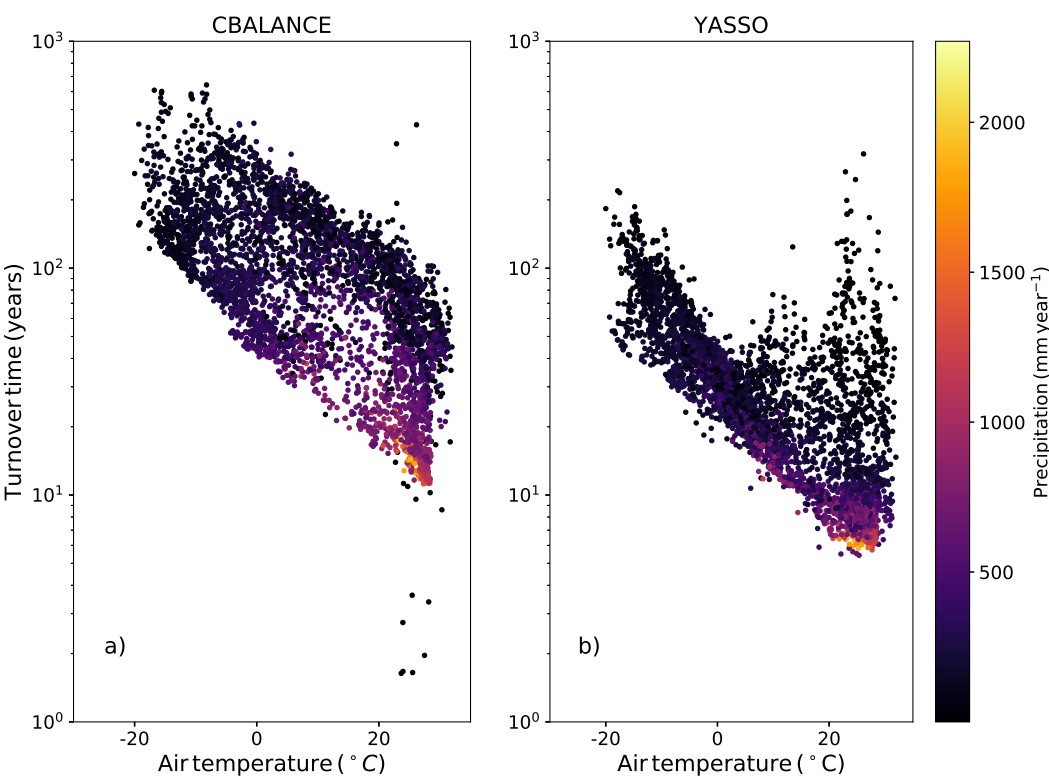

**Figure 4.** The turnover times at different temperature and precipitation regimes for the CBALANCE (a) and YASSO (b) models.

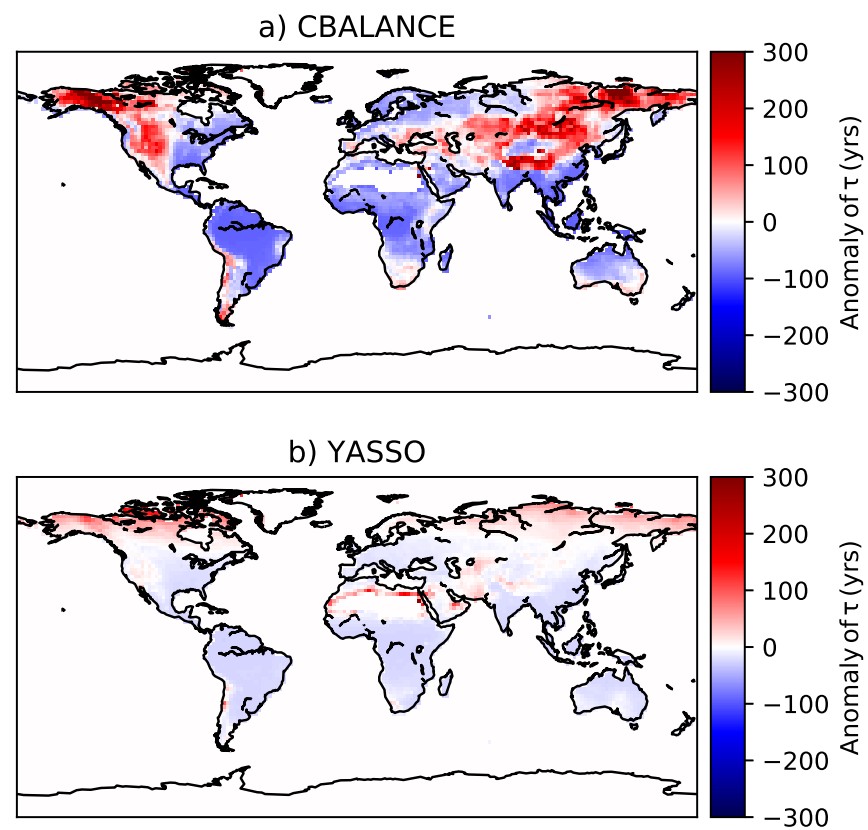

**Figure 5.** Turnover time ($\tau$) anomalies for CBALANCE (a) and YASSO (b). The average turnover time that was subtracted was 104 years for CBALANCE and 31 years for YASSO.

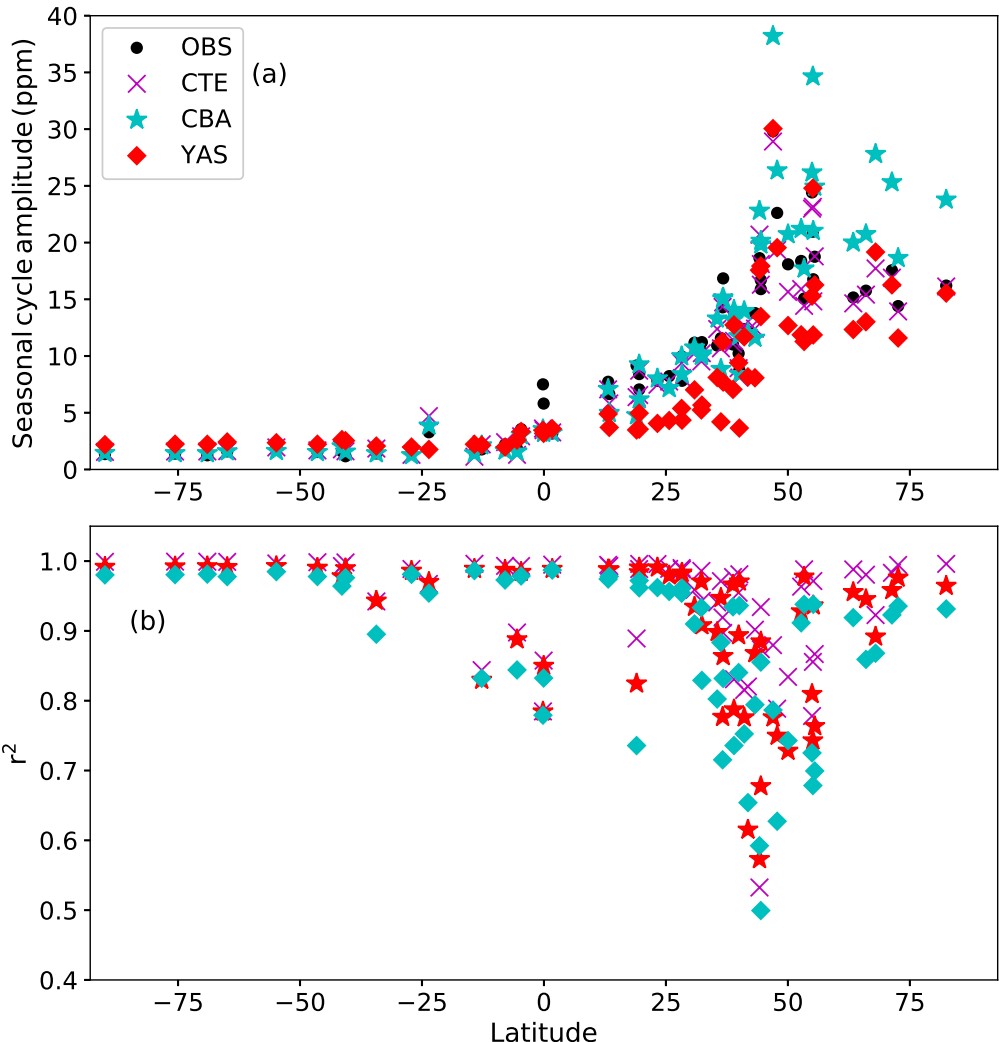

**Figure 6.** The seasonal cycle amplitudes of atmospheric $CO_2$ in ppm (a) and $r^2$ (b) between the simulations and observations at different Global Atmosphere Watch stations as a function of latitude. The black circles denote observations, the magenta crosses are the results from the CarbonTracker Europe 2016 (CTE), the cyan stars are the results from the CBALANCE (CBA) run and the red diamonds are the results from the YASSO (YAS) run.

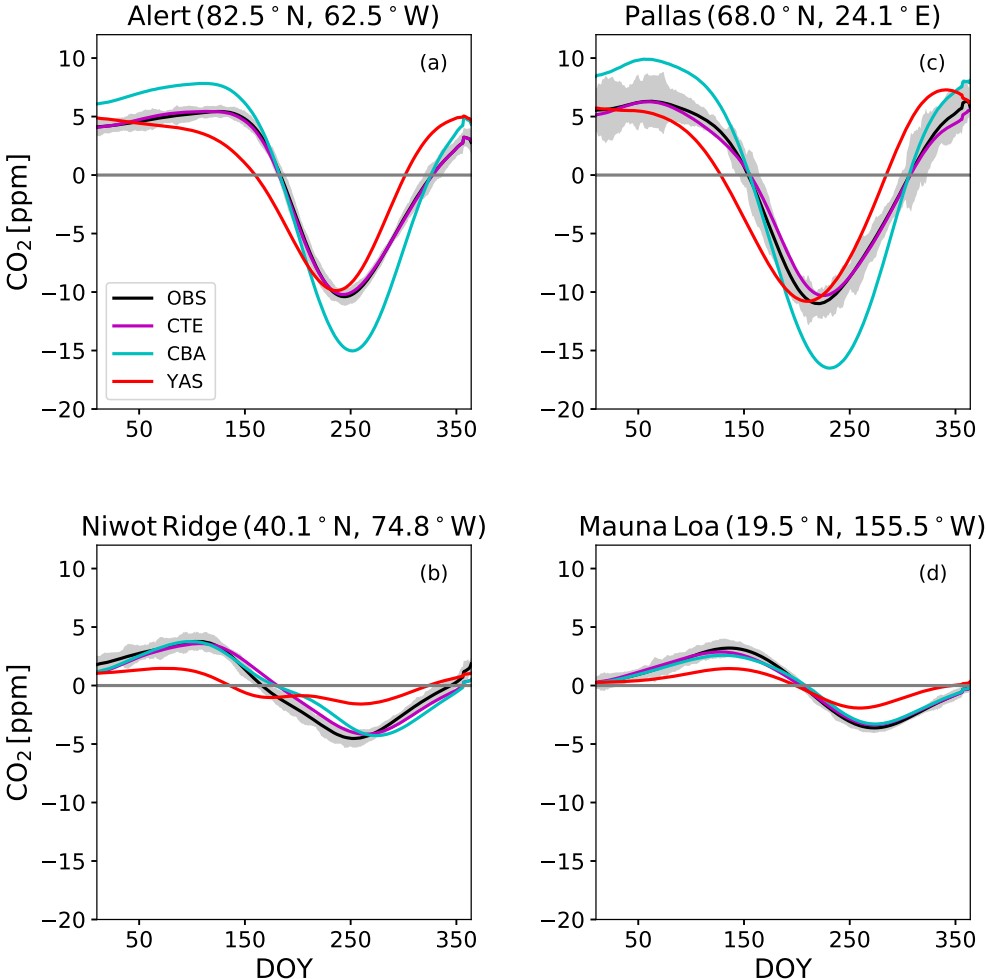

**Figure 7.** The detrended seasonal cycles of atmospheric $CO_2$ at four Global Atmospheric Watch sites: Alert (a), Pallas (c), Niwot Ridge (b), and Mauna Loa (d) for observations (OBS) in black, CarbonTracker Europe 2016 (CTE) in magenta, and the two JSBACH model version with CBALANCE (CBA) in cyan line and YASSO (YAS) in red line. The solid grey line denotes the zero line. The grey shaded area is showing the standard deviations of the observations after detrending.

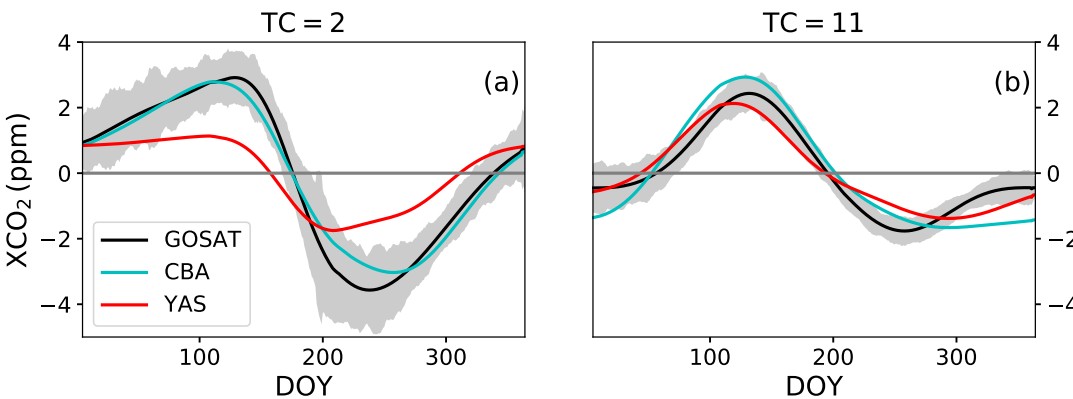

**Figure 8.** The seasonal cycles of detrended atmospheric $XCO_2$ at TransCom regions two, southern part of North America (a) and eleven, Europe (b). The grey line is showing the standard deviations of the observations after detrending. The observations are in black, CBALANCE (CBA) model results in cyan and YASSO (YAS) model results in red. The solid grey line denotes the zero line.

**Table 1.** Global C storage in the two different model formulations averaged over 2001-2014. For the YAS model the eight above ground pools are summed to obtain the litter pool, while the remaining 10 pools (below ground and humus) represent the soil pool.

| C pool (Pg C) | CBA | YAS |
|---|---|---|
| Litter C | 171 | 263 |
| Soil C | 3217 | 703 |
| Vegetation C | 432 | 432 |

**Table 2.** Global terrestrial C fluxes from the two different model formulations averaged over 2001-2014.

| Row | Flux ($PgCyr^{-1}$) | CBA | YAS |
|-----|------------------------|-------|-------|
| A | Net $CO_2$ flux (A = -B + E + G + H + I + J) | -1.68 | -1.75 |
| B | GPP | 167 | same |
| C | Heterotrophic resp. $R_h$ | 66.1 | 65.5 |
| D | Autotrophic resp. $R_a$ | 89.9 | same |
| E | TER (E = C + D) | 156 | 155 |
| F | NPP (F = B - D) | 77.4 | same |
| G | Direct land cover change | 2.30 | same |
| H | Fire | 1.60 | 2.10 |
| I | Harvest | 0.23 | same |
| J | Herbivory | 5.54 | same |

**Table 3.** The Pearson correlation $r$ values for the different latitudinal zones between modelled heterotrophic respiration and the environmental drivers of the CBALANCE (CBA) and YASSO (YAS) models. The environmental drivers are all calculated as monthly means for the latitudinal zones. Significant correlation (p-value $< 0.05$) have been written in bold. $\alpha$ is the relative soil moisture, $T_{soil}$ and $T_{air}$ are soil and air temperature, and $P_a$ is the precipitation.

| Lat. zone | CBA vs. $\alpha$ | CBA vs. $T_{soil}$ | YAS vs. $P_a$ | YAS vs. $T_{air}$ | YAS vs. $\alpha$ |
|---|---|---|---|---|---|
| 60°N -90°N | -0.22 | **0.96** | **0.95** | **0.90** | -0.48 |
| 30°N -60°N | **−0.81** | **0.99** | **0.98** | **0.95** | -0.92 |
| 0°N -30°N | **0.96** | 0.49 | **0.96** | **0.93** | 0.58 |
| 0°S -10°S | **0.92** | 0.03 | **0.93** | 0.52 | 0.46 |
| 10°S -30°S | **0.94** | 0.38 | **0.93** | **0.92** | 0.48 |
| 30°S -60°S | -0.46 | **0.76** | **0.78** | **0.95** | **−0.91** |

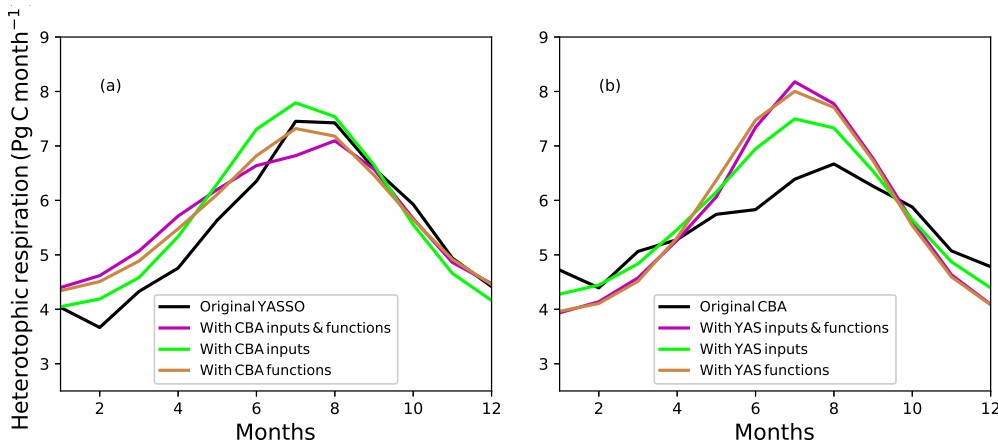

**Figure A1.** Different annual cycles of the heterotrophic respiration ($R_h$) predicted by the YASSO (YAS) (a) and CBALANCE (CBA) (b) model and the different alternatives from the box model calculation.

**Table A1.** The amplitude of global heterotrophic respiration within a year in different box model formulations. The input variables or functions that differ from the original model formulation are in bold letters.

| Line | Option | Amplitude $(\mathrm{PgCyr^{-1}})$ |
|---|---|---|
| A) | YAS - Original model | 3.8 |
| B) | YAS with inputs $\mathbf{T_{soil}}$ and $\alpha$ and functions $\mathbf{f_{CBA,T_{soil}}}$ and $\mathbf{f_{CBA,\alpha}}$ | 2.7 |
| C) | YAS with inputs $\mathbf{T_{soil}}$ and $\alpha$ and functions $f_{YAS,T_{air}}$ and $f_{YAS,P_a}$ | 3.7 |
| D) | YAS with inputs $T_{air}$ and $P_a$ and functions $\mathbf{f_{CBA,T_{soil}}}$ and $\mathbf{f_{CBA,\alpha}}$ | 3.0 |
| A) | CBA - Original model | 2.3 |
| B) | CBA with inputs $\mathbf{T_{air}}$ and $\mathbf{P_a}$ and functions $\mathbf{f_{YAS,T_{air}}}$ and $\mathbf{f_{YAS,P_a}}$ | 4.2 |
| C) | CBA with inputs $\mathbf{T_{air}}$ and $\mathbf{P_a}$ and functions $f_{CBA,T_{soil}}$ and $f_{CBA,\alpha}$ | 3.2 |
| D) | CBA with inputs $T_{soil}$ and $\alpha$ and functions $\mathbf{f_{YAS,T_{air}}}$ and $\mathbf{f_{YAS,P_a}}$ | 4.0 |