# Peer review of "Evaluating two soil carbon models within a global land surface model using surface and spaceborne observations of atmospheric CO2 mole fractions"

_Biogeosciences, 2020_

## Referee Comment (RC1) · Anonymous Referee #1 · 24 Mar 2020

Thum and colleagues present an interesting study looking at how different soil models influence atmospheric CO2 fluxes, which can be compared with ground and space-based observations. The study is thorough and well written, but in my estimation, it somewhat glosses over some of the caveats that should likely be considered with trying to evaluate component fluxes (here HR), with atmospheric CO2 observations that serve as a proxy for NEE.

Major concerns: Introduction, I found the basic primer on the global carbon cycle a little pedantic at times, with phrases like "Photosynthesis takes place in the green plant
parts". As well as wandering, combining information on soil respiration and soil C stocks without really developing these ideas in a focused way that supports the direction or intent of the paper. I realize this is more stylistic that substantiative comment, but would encourage the authors to revise the introduction with focused topic sentences for each paragraph that introduces background literature and develop ideas in a way that focuses attention on the work to be presented.

I'm trying to wrap my head around how biases in the magnitude (and timing) of GPP simulated by JSBACH confound results presented here and I fundamentally disagree with the statement on Line 392. • Should SI4 and Fig 2 be combined into one display item in the main text, using the same y-axis units for both? • Could you show the annual cycle of NPP or HR globally, in addition to the latitudinal bins shows in SI2? [ maybe this goes into SI ]. • Would similar plots of regional & global AR or NPP also be helpful (in SI)? • It's not really clear that the magnitude seasonal cycle of HR with YAS is so large in mid latitudes (30-60N; Fig 4), and is of equal and nearly opposite magnitude to the high GPP biases in this region, resulting in a lower than observed NEE (Fig 6-7). • I'm wondering how any version of JSBACH captures appropriate seasonal peak to trough dynamics of NEE Fig 6 given biases in the magnitude of GPP fluxes? This means that the timing and or magnitude of AR or HR fluxes must be compensating to generate reasonable seasonal cycle of NEE. Given known biases in the seasonal cycles of GPP (Fig S2), what modifications are needed to improve the representation of plant and soil dynamics in JSBACH? • In summary, could one erroneously chose an inferior soil biogeochemical model that give 'better' NEE fluxes with atmospheric observations, but that's fundamentally just masking over / compensating biases in GPP? This is, 'getting the right answer for the wrong reasons?

Throughout on display items, the shades of grey make it kind of hard to distinguish models and observations. This is especially true for Figs 5-7). Is there any harm in using colors?

Minor and technical concerns: There are enough abbreviations in the text that it somewhat distracts from the readability of the manuscript. I'd encourage removing some of these less standard ones if possible (e.g. SCA, GAW).

There are also several very short paragraphs (even just one sentence long), that should either be further developed or merged into related paragraphs.

L19: the IAV of fluxes are never communicated here (although they could be easily brought into text and display items.

L26: Maybe remove 'natural'

L39: I might include van Gestel et al's 2018 critique of the Crowther paper to make this point.

L56: The connection between benchmarking global soil C models (the topic of the last paragraph) and global CO2 flux observations is a little rocky and unclear. Reading between the lines, I think it's a very good idea- but the connections about how / why it may be considered a useful way to evaluate soil biogeochemical models should be clarified. That said, others have recently used a similar approach (Basile et al 2020), which could be useful in contextualizing the introduction of the present work.

I'm also assuming the authors will discuss some of the assumptions being made in evaluating HR fluxes with atmospheric CO2 observations that may include biases in the timing and magnitude of GPP and AR fluxes from (JSBACH), potential errors imparted by the atmospheric transport model (TM5), or challenges in interpreting total column CO2 observations- especially from space. Maybe it's worth foreshadowing some of these in the introduction? See also refs from Keppel-Aleks et al (below).

Line 67: this sentence seems awkwardly phrased, maybe drop 'in' and 'far'.

Line 84: Is there reference for TM5, or example of where / how it's been used?

Line 97: Randerson et al 1997 found that the timing of litterfall was important for controlling the timing of HR fluxes. Is the same true in JSBACH? How well does the model

simulate this phenology?

Line 140, I realize it's likely in your previous publications, but is it worth noting how litter chemistry from JSBACH PFTs is translated onto the YASSO litter quality definitions? The way this is presented in kind of confusing & disconnected

Line 143 define PFT

Line 159 constraints for what?

Line 175 is 'atmospheric' redundant here?

Line 205: Should this be S2.

Line 250: Why run statics on YAS fluxes and alpha, when the model uses precipitation to moderate decomposition rates (eq5)?

Are the correlations between environmental drivers and HR fluxes really that surprising or interesting (Table 3, Line 241-254 & 366)? The models have these assumptions apriori (Methods, Fig. S1). In places with large seasonal variation in soil or air temperature (arctic), temperature is important control over seasonal HR fluxes. In places with little annual variation in temperature (tropics) moisture is a more important control. I'm not really sure what readers are supposed to learn from this analysis?

Fig 6: Dotter is not a word.

Fig 6 & 7, can uncertainty estimate (or interannual variability) be shown for observations?

Table 3, why not just report r values, so negative correlations can be more clearly illustrated. Can't statistically significant correlations be highlighted (not just results with high r2)?

L345, I'm not sure better agreement with CMIP5 models is necessarily a good thing, based on Kathe's work. Moreover, the calculation of global turnover times seems to

mask important regional patterns. Instead, see Koven et al. 2017.

L395: where are results supporting these claims shown?

L405: This doesn't seem like a standalone paragraph, nor is it really clear how it relates to the results being discussed regarding carbon tracker and JSBACH land C uptake.

L408: what biases are not important here?

L414: while I agree with this statement, however it's not done in the work presented. Multiple benchmarks, however, could be presented, again see Koven et al. 2017, or Todd-Brown's work that's already cited). Indeed it seems for a flux based analysis like this some more rigorous evaluation of upstream fluxes (e.g. GPP) is pretty important.

References: Basile, S. J., Lin, X., Wieder, W. R., Hartman, M. D., and Keppel-Aleks, G.: Leveraging the signature of heterotrophic respiration on atmospheric CO2 for model benchmarking, Biogeosciences, 17, 1293–1308, https://doi.org/10.5194/bg-17-1293-2020, 2020.

van Gestel, N., Shi, Z., van Groenigen, K. J., Osenberg, C. W., Andresen, L. C., Dukes, J. S., et al. (2018). Predicting soil carbon loss with warming. Nature, 554(7693), E4–E5. https://doi.org/10.1038/nature25745

Keppel-Aleks, G., Randerson, J. T., Lindsay, K., Stephens, B. B., Keith Moore, J., Doney, S. C., ... & Tans, P. P. (2013). Atmospheric carbon dioxide variability in the Community Earth System Model: Evaluation and transient dynamics during the twentieth and twenty-first centuries. Journal of Climate, 26(13), 4447-4475.

Keppel-Aleks, G., Wennberg, P. O., & Schneider, T. (2011). Sources of variations in total column carbon dioxide. Atmospheric Chemistry and Physics, 11(8), 3581-3593

Koven, C. D., Hugelius, G., Lawrence, D. M., & Wieder, W. R. (2017). Higher climatological temperature sensitivity of soil carbon in cold than warm climates. Nature Climate Change, 7(11), 817-822. doi:10.1038/nclimate3421

Randerson, J. T., Thompson, M. V., Conway, T. J., Fung, I. Y., and Field, C. B.: The contribution of sources and sinks to trends in the seasonal cycle of atmospheric carbon dioxide, Global Biogeochem. Cy., 11, 535–560, https://doi.org/10.1029/97GB02268, 1997.

———————————————————————

---

## Referee Comment (RC2) · Anonymous Referee #2 · 2 Apr 2020

Thum et al. present an interesting approach for the evaluation of soil carbon models in a land surface model by using atmospheric CO2 observations. I like the basic idea of the study and the work is methodological sound. However, I actually got bored and disappointed when I was reading the paper. This is too a large degree caused by the presentation (text and figures) of the material (I agree with all the points by reviewer 1):

1) It is not clear what the purpose of the study is. Only in the discussion it's written that the "aim was to use atmospheric observations to benchmark soil carbon models". If this was the aim, the evaluation of just two modules of JSBACH is insufficient (and causes

my disappointment). It would be better to clearly state already in the introduction that the aim was to evaluate two soil modules of JSBACH. However, if this is the case, the manuscript would do better as a model evaluation paper in Geoscientific Model Development. Generally, the text is written as a model evaluation study and I don't find important results for the general Biogeosciences. My feeling is that the paper should go beyond a model evaluation and include some more substantial scientific questions, hypotheses and findings in order to fit into Biogeosciences.

2) If the "aim was to use atmospheric observations to benchmark soil carbon models" (as stated), I would expect a more detailed description of the assumptions and a detailed analysis on how to use atmospheric $CO_2$ observations for the benchmarking, including how to disentangle the contributions of GPP and Reco on $CO_2$, the role of uncertainties in observations and atmospheric transport, and how different regions contribute to the $CO_2$ seasonality. Especially the later points would help to potentially benchmark soil carbon simulations if different parts of the world.

3) As already stated by reviewer 1, the text needs substantial rewrite. The text has no clear structure, topic sentences are missing, some chapters are too long (especially 1 and 3.1). For example, the first section of the results (3.1) report mainly minor results (including references to the supplement) but does not report the most important results. In addition, I recommend to split this section in further sub-chapters to improve the structure of the text.

4) Figures: I'm sorry, but reading the figures in the main text and in the supplement was a nightmare! The figures are too small and the grey colours make it almost impossible to distinguish the different model runs and observations. Please improve all figures.

5) The discussion of GPP is over-simplistic. JSBACH overestimates GPP and has in some regions shifts in the seasonality. Hence it remains unclear which soil carbon model is the better one because the comparison of $CO_2$ seasonality is also affected by wrong simulations of GPP. Could it be an option to force more realistic GPP estimates

into JSBACH or mix GPP from data-driven estimates with Rh from JSBACH in TM5?

6) Please describe if permafrost was simulated in the JSBACH runs and how the simulation or non-simulation of permafrost contributed to soil carbon simulations.

- Figure 1: Explain the numbers in the caption

- Figure 2: Even if the soil carbon stocks have been already evaluated, it would be still helpful to add 1 or 2 maps from an observation-based product for comparison.

- Table 2: There seems to be a mistake in the results for TER, as those can't be the same numbers.

---

## Author Comment (AC1) · 30 Apr 2020

Reply to reviewer #1 by Thum et al. (The original comments by the reviewer are in violet and the replies by the authors are in black.)

Thum and colleagues present an interesting study looking at how different soil models influence atmospheric CO2 fluxes, which can be compared with ground and space-based observations. The study is thorough and well written, but in my estimation,it somewhat glosses over some of the caveats that should likely be considered with trying to evaluate component fluxes (here HR), with atmospheric CO2 observations that serve as a proxy for NEE.

We thank the reviewer for these in-depth comments and views. We hope we are able to address the concerns the reviewer is bringing up in our replies and in the revised version of the manuscript in a satisfactory manner.

Major concerns: Introduction, I found the basic primer on the global carbon cycle a little pedantic at times, with phrases like "Photosynthesis takes place in the green plant parts". As well as wandering, combining information on soil respiration and soil C stocks without really developing these ideas in a focused way that supports the direction or intent of the paper. I realize this is more stylistic that substantiative comment, but would encourage the authors to revise the introduction with focused topic sentences for each paragraph that introduces background literature and develop ideas in a way that focuses attention on the work to be presented.

We thank the reviewer for this advice in improving the introduction. We will re-write the introduction to be less pedantic and also take on the other good hints forward provided here. The reason we wrote the sentence "*Photosynthesis takes place in the green plant parts*" this way was to highlight the ending of this sentence: that the remote sensing is therefore able to detect the terrestrial photosynthesis.

I'm trying to wrap my head around how biases in the magnitude (and timing) of GPP simulated by JSBACH confound results presented here

In the JSBACH simulations the biosphere has been first spun-up to steady state in 1860 and the current land sink is resulting from the simulation period between 1860 to present. In 1860 the global net land $CO_2$ balance is therefore zero, and if the gross primary production is then overestimated, the autotrophic and heterotrophic respiration are then also overestimated.

and I fundamentally disagree with the statement on Line 392.

The sentence in line 392 is: "*The biases between $XCO_2$ from satellite retrievals and the model results originating from the JSBACH simulations are relatively large and this is likely caused by the use of a posteriori ocean fluxes from the CTE2016.*"

The biases we talk about in the sentence in line 392 is connected to the absolute differences in the $CO_2$ concentration values, that were reported in lines 294 and 322 [we should have mentioned also the site level observations, and not only the satellite observations in this sentence].

This bias we mean here is not connected to the sink or source terms separately but to the net land sink, that we report in Table 2 to be -1.68 PgC for CBA and -1.75 for YAS, and how it differs from the net land sink from the CTE2016 framework. With Fig. S10 we aimed to demonstrate how development of the bias in the absolute $CO_2$ values is in line with these different net land sink estimates, as we have explained in lines 394-397.

Re-writing the sentence in line 392 requires emphasizing better which bias we are meaning. We agree with the reviewer, that within the JSBACH model the modelled GPP is overestimated and the timing doesn't match perfectly. We will bring this up more in the new version of the manuscript, adding that also the bias of GPP influences the modelling skill of the system and not only the performance of the soil carbon model itself.

Should SI4 and Fig 2 be combined into one display item in the main text, using the same y-axis units for both?

The Fig. 2 displays a soil carbon map and Fig. S4 the anomalies from the turnover rates. If we put the color code to be the same for the both models, the spatial patterns won't be anymore visible. Since the reviewer 2 requested to also show observation based soil carbon maps, we will add observations to Fig.2 and still keep Fig. S4 and Fig. 2 separate.

Could you show the annual cycle of NPP or HR globally, in addition to the latitudinal bins shows in SI2? [ maybe this goes into SI ].
Would similar plots of regional & global AR or NPP also be helpful (in SI)?

We will present a plot of NPP, heterotrophic respiration (HR) and NEE, both globally as well as divided into latitudinal bins. Additionally we add a plot of autotrophic respiration (AR) to the supplement, showing both the global and latitudinally divided yearly cycles.

It's not really clear that the magnitude seasonal cycle of HR with YAS is so large in mid latitudes (30-60N; Fig 4), and is of equal and nearly opposite magnitude to the high GPP biases in this region, resulting in a lower than observed NEE (Fig 6-7).

Magnitude of the seasonal cycle at these latitudes by YAS is 2.5 Pg month$^{-1}$, and the GPP bias in this region is approximately 0.05 PgC day$^{-1}$ = 1.5 PgC month$^{-1}$. How the difference between the heterotrophic respiration from the YAS and CBA models influence the global and latitudinally segregated NEE values will be shown in a figure in the revised manuscript.

I'm wondering how any version of JSBACH captures appropriate seasonal peak to trough dynamics of NEE Fig 6 given biases in the magnitude of GPP fluxes? This means that the timing and or magnitude of AR or HR fluxes must be compensating to generate reasonable seasonal cycle of NEE.

The annual magnitudes of the autotrophic and heterotrophic respiration have been given in Table 2 in the first version. The autotrophic respiration is relatively large. In the revised version of the manuscript we will show the annual cycles of NPP and NEE globally and in latitudinal bins and autotrophic respiration similarly in the supplementary material. The compensation by the autotrophic and heterotrophic respirations become visible in these plots and we will also discuss these plots in the revised manuscript.

Given known biases in the seasonal cycles of GPP (Fig S2), what modifications are needed to improve the representation of plant and soil dynamics in JSBACH?

In the data assimilation study by Castro-Morales et al. (2019), where satellite observed FAPAR and atmospheric $CO_2$ molar fractions were used in the assimilation, the maximum LAI value in the tropics was lowered via optimization. This itself will not enhance the seasonal cycle in the tropics, but would bring its absolute value down. To improve the seasonality of the tropical GPP, re-parametrization of the phenology model for the tropical plant functional type would likely help.

The timing is not so much off in the region 30ºN - 60ºN, but for the phenology during senescence period there might be room for improvement. North of 60º the modelled GPP is peaking too late, but the start of the growing season is occurring at the same time with the FLUXCOM results.

A study done for Finland, comparing growing season onset by satellite observations to JSBACH simulation output found out that it was enough to improve the onset of the deciduous forest by re-parameterizing the phenology module, however, to improve the onset of the coniferous forest it was necessary to add seasonality to the temperature responses of the photosynthesis parameters (Böttcher et al., 2016). It is therefore not so straightforward to say, which changes would improve the seasonal cycle of GPP. Improving the phenology cycle might be a step forward, but in the tropics how the plants experience dry conditions might be also off.

For the soil dynamics, based on the results of this study, we strongly recommend moving towards using the soil moisture as a driver for the YAS model. This was already mentioned in the earlier manuscript version.

In summary,could one erroneously chose an inferior soil biogeochemical model that give 'better' NEE fluxes with atmospheric observations, but that's fundamentally just masking over /compensating biases in GPP? This is, 'getting the right answer for the wrong reasons?

The reviewer is right, that the biased GPP might influence the results. Therefore we will modify the wordings of this manuscript so that we will not talk about benchmarking, but will emphasize that the aim was to see how well we can try to assess the differences between the soil carbon formulations within this kind of system and acknowledge that when it comes to ranking the models, we'd need a more data-driven system such as testbed, where the soil carbon models can be forced with certain plant productivity inputs (Wieder et al., 2018) or a systematic assessment of several variables against observations. The aim of this study was comparison of the two soil carbon models, which both had the same GPP input, and while the GPP bias compared to observations does have implications for numerical benchmarking, it does not take away the importance and conclusions of this work.

While the GPP of the JSBACH is not a perfect match to observations, it is the same in the both model simulations and the aim was to evaluate the soil carbon modules within a global land surface model, that includes several different process descriptions, e.g. also fires and land use changes. JSBACH is also part of the Earth System Model of the Max Planck Institute and an IPCC model, and it is state-of-the-art land surface model.

Throughout on display items, the shades of grey make it kind of hard to distinguish models and observations. This is especially true for Figs 5-7). Is there any harm in using colors?

We will add colors to all of the figures that were black and white in the earlier version.

Minor and technical concerns: There are enough abbreviations in the text that it some what distracts from the readability of the manuscript. I'd encourage removing some of these less standard ones if possible (e.g. SCA, GAW).There are also several very short paragraphs (even just one sentence long), that should either be further developed or merged into related paragraphs.

We will use less abbreviations in the second version of the manuscript and pay attention to the length of the paragraphs.

L19: the IAV of fluxes are never communicated here (although they could be easily brought into text and display items.

We have added the annual values to the plots that show seasonal cycles of NPP, TER and NEE.

L26: Maybe remove 'natural'

Removed.

L39: I might include van Gestel et al's 2018 critique of the Crowther paper to make this point.

Thank you for bringing up this point, we will do this.

L56: The connection between benchmarking global soil C models (the topic of the lastparagraph) and global CO2 flux observations is a little rocky and unclear. Reading between the lines, I think it's a very good idea- but the connections about how / why it may be considered a useful way to evaluate soil biogeochemical models should be clarified. That said, others have recently used a similar approach (Basile et al 2020), which could be useful in contextualizing the introduction of the present work.

Thanks, we will use this paper in bringing this work better to the context and discuss better the link between the connection of atmospheric $CO_2$ molar fractions and soil model evaluation.

I'm also assuming the authors will discuss some of the assumptions being made in evaluating HR fluxes with atmospheric CO2 observations that may include biases in the timing and magnitude of GPP and AR fluxes from (JSBACH), potential errors imparted by the atmospheric transport model (TM5), or challenges in interpreting total column CO2 observations- especially from space. Maybe it's worth foreshadowing some of these in the introduction?

We thank the reviewer for this insight, and find the idea of bringing them up in the introduction a good idea. We will do this.

See also refs from Keppel-Aleks et al (below).

We also thank the reviewer for pointing to these useful references that we'll make use of in the revised version of the manuscript.

Line 67: this sentence seems awkwardly phrased, maybe drop 'in' and 'far'.

We'll rephrase this sentence.

Line 84: Is there reference for TM5, or example of where / how it's been used?

In the earlier version we had added more information about TM5 only in section 2.3. We will in the new version add some references already here in the introduction.

Line 97: Randerson et al 1997 found that the timing of litterfall was important for controlling the timing of HR fluxes. Is the same true in JSBACH? How well does the model simulate this phenology?

As shown in Table 3, the decomposition is for a large part regulated by the environmental conditions. For completeness, we now checked the correlations between litterfall and heterotrophic respiration (HR) similarly as we did for the environmental drivers in Table 2: There is a positive significant correlation between the litterfall and heterotrophic respiration in region 30 N - 60 N with

both of the model versions. The increase of heterotrophic respiration is anyhow preceding the litter flux. We will add a plot of this to the supplement and discuss this in the manuscript.

Interesting point in the study by Randerson et al is that changing litter quality would be making changes to the decomposition they are seeing in their study. This is something that we could actually test with the YAS model and we will also mention this in the new version of the manuscript.

Line 140, I realize it's likely in your previous publications, but is it worth noting how litter chemistry from JSBACH PFTs is translated onto the YASSO litter quality definitions? The way this is presented in kind of confusing & disconnected

We will clarify this in the revision of the manuscript. The division of the incoming litter from the JSBACH model per PFT is based on observations from different ecosystems (Trofymow et al., 1998; Berg et al., 1991a, 1991b; Gholz et al., 2000). We will add this information to the new version of the manuscript.

Line 143 define PFT

Thank you, we provide a definition for it here.

Line 159 constraints for what?

This comment refers to the sentence: '*The 3-hourly meteorological fields from ECMWF ERA-Interim (Dee et al., 2011) were used as constraints.*' We will modify this sentence to say that the TM5 model is run with the 3-hourly meteorological data from ERA-Interim.

Line 175 is 'atmospheric' redundant here?

Thanks, removed.

Line 205: Should this be S2.

Thanks, corrected.

Line 250: Why run statics on YAS fluxes and alpha, when the model uses precipitation to moderate decomposition rates (eq5)?

As explained in the manuscript in line 378, the precipitation is used as a driver in YAS, since it has been considered to estimate the soil moisture. While this might be true at the annual timescales, in which this model has been originally developed, this comparison here in Table 3 shows that it is not justified at monthly scale. We have commented this in the earlier manuscript version in line lines 384-386 with: "*Precipitation has been originally used in the YAS model as a proxy for soil moisture, since enough accurate soil moisture observations for model development haven't been available. Clearly, this idea needs reconsideration as our results show that at zonal spatial scales and monthly temporal scale the Rh from YAS is not at all correlated to soil moisture variable $\alpha$.*" in the Discussion and in line 423-424 in the Conclusions: "*This suggests that use of precipitation as a proxy for soil moisture might not be sensible in sub-annual time scales.*"

Are the correlations between environmental drivers and HR fluxes really that surprising or interesting (Table 3, Line 241-254 & 366)? The models have these assumptions a priori (Methods, Fig. S1). In places with large seasonal variation in soil or air temperature (arctic), temperature is

important control over seasonal HR fluxes. In places with little annual variation in temperature (tropics) moisture is a more important control. I'm not really sure what readers are supposed to learn from this analysis?

While we do have these clear response functions that are driving the soil carbon models, it is not necessarily clear at which part of the response we're in these different ecosystems. E.g. in the tropical zone, area 10ºS - 0º in Fig. S7, the YAS model is actually reaching the saturation level in respect to moisture.

And as the reviewer later mentions, the litter flux could play a role, but this is not a role in our model and these high correlation values to the environmental drivers already suggest that. These correlations also function as comparison for the soil moisture vs. YAS Rh, as mentioned.

Fig 6: Dotter is not a word.

Thanks, it was a typo. In the revised version we will have colors in this figure and not have a dotted line there.

Fig 6 & 7, can uncertainty estimate (or interannual variability) be shown for observations?

Thanks for the suggestion, we will add interannual variability of the seasonal cycle amplitude to the plots.

Table 3, why not just report r values, so negative correlations can be more clearly illustrated. Can't statistically significant correlations be highlighted (not just results withhigh r2)?

Thanks for this suggestion, we will do so.

L345, I'm not sure better agreement with CMIP5 models is necessarily a good thing,based on Kathe's work. Moreover, the calculation of global turnover times seems to mask important regional patterns. Instead, see Koven et al. 2017.

We considered to add this metric here, since it has been used, but this was more to complement the Figure S4 that was showing a map of the turnover rate anomalies by the two models. We agree with the reviewer that a global value does not include many important features and add here also a plot similar to Koven et al. 2017, where the turnover rates are shown as a function of air temperature and the precipitation is visible via coloring.

L395: where are results supporting these claims shown?

The sentence in this line is: '*The global land sink of JSBACH is approximately -1.7 PgCyr −1 (Table 2) and therefore the used ocean fluxes cause a bias to the simulated atmospheric $CO_2$ molar fraction.*'

We used posteriori ocean fluxes from the CTE2016 in this study. These ocean fluxes have been optimized using the terrestrial carbon cycle of the CTE2016, the SiBCASA-GFED4 model (van der Velde et al., 2014), and the fire emission fluxes that have been estimated from satellite observed burned area (Giglio et al., 2013). The fossil fuel emissions have not been optimized.

Since the optimization has been done with this other set of terrestrial carbon fluxes, this would be the likely cause for the bias. The fire fluxes of the CTE2016 are also different to JSBACH, so that

could be another source for the discrepancy. We will modify this paragraph to add this point and also to clarify why we consider the ocean fluxes to be causing this bias.

L405: This doesn't seem like a standalone paragraph, nor is it really clear how it relates to the results being discussed regarding carbon tracker and JSBACH land C uptake.

Thanks for noting this, we will tie this information to the other paragraphs.

L408: what biases are not important here?

We admit that this line is unclear. We were referring here to the biases in the absolute $CO_2$ molar fraction values, and that these are not important, since the differences in the absolute values are not playing a role, since we concentrated in this analysis on the seasonal cycles and only mentioned the bias in the absolute $CO_2$ for completeness. We will rephrase this sentence.

L414:  while I agree with this statement, however it's not done in the work presented. Multiple benchmarks, however, could be presented, again see Koven et al.  2017, or Todd-Brown's work that's already cited). Indeed it seems for a flux based analysis like this some more rigorous evaluation of upstream fluxes (e.g. GPP) is pretty important.

The sentence on the line is: "*We demonstrated how atmospheric $CO_2$ observations can be used to benchmark soil carbon models and that it is important to benchmark models across several different variables.*" We will remove the word benchmarking from the revised version of the manuscript, since we did not perform rigorous numerical benchmarking in this study, but were evaluating two different soil carbon models within one global land surface model within the constraints given by that system.

**References:**

Berg, B., et al. (1991a), Data on needle litter decomposition and soil climate as well as site characteristics for some coniferous forest sites. Part I. Site characteristics, Rep. 41, Dep. of Ecol. and Environ. Res., Swed. Univ. of Agric. Sci., Uppsala, Sweden.

Berg, B., et al. (1991b), Data on needle litter decomposition and soil climate as well as site characteristics for some coniferous forest sites. Part II. Decomposition data, Rep. 42, Dep. of Ecol. and Environ. Res., Swed. Univ. of Agric. Sci., Uppsala, Sweden.

Böttcher, K., Markkanen, T., Thum, T., Aalto, T., Aurela, M., Reick, C.H., Kolari, P., Arslan, A.N. and J. Pulliainen 2016. Evaluating Biosphere Model Estimates of the Start of the Vegetation Active Season in Boreal Forests by Satellite Observations. Remote Sensing, 8, 580.

Castro-Morales, K., Schürmann, G., Köstler, C., Rödenbeck, C., Heimann, M., and Zaehle, S.: Three decades of simulated global terrestrial carbon fluxes from a data assimilation system confronted with different periods of observations, Biogeosciences, 16, 3009–3032, https://doi.org/10.5194/bg-16-3009-2019, 2019.

Dalmonech, D. and Zaehle, S.: Towards a more objective evaluation of modelled land-carbon trends using atmospheric $CO_2$ and satellite-based vegetation activity observations, Biogeosciences, 10, 4189–4210, https://doi.org/10.5194/bg-10-4189-2013, 2013.

Gholz, H. L., D. A. Wedin, S. M. Smitherman, M. E. Harmon, and W. J. Parton (2000), Long   term dynamics of pine and hardwood litter in contrasting environments: toward a global model of

decomposition, Global Change Biol., 6, 751–765, doi:10.1046/j.1365-2486.2000.00349.x.

Giglio, L., Randerson, J. T., and van der Werf, G. R.: Analysis of daily, monthly, and annual burned area using the fourth-generation global fire emissions database (GFED4), J. Geophys. Res.-Biogeo., 118, 317–328, https://doi.org/10.1002/jgrg.20042, 2013.

Trofymow, J. A., et al. (1998), The Canadian Intersite Decomposition Experiment (CIDET): Project and site establishment report, Inf. Rep. BC-X-378, Pac. For. Cent., Victoria, B. C., Canada.

van der Velde, I. R., Miller, J. B., Schaefer, K., van der Werf, G. R., Krol, M. C., and Peters, W.: Terrestrial cycling of 13 CO2 by photosynthesis, respiration, and biomass burning in SiBCASA, Biogeosciences, 11, 6553–6571, https://doi.org/10.5194/bg-11-6553-2014, 2014.

Wieder, WR, Hartman, MD, Sulman, BN, Wang, Y  P, Koven, CD, Bonan, GB. Carbon cycle confidence and uncertainty: Exploring variation among soil biogeochemical models. *Glob Change Biol*. 2018; 24: 1563– 1579. https://doi.org/10.1111/gcb.13979

---

## Author Comment (AC2) · 30 Apr 2020

Reply to reviewer #2 by Thum et al. (The original comments by the reviewer are in violet and the replies by the authors are in black.)

Thum et al. present an interesting approach for the evaluation of soil carbon models in a land surface model by using atmospheric CO2 observations. I like the basic idea of the study and the work is methodological sound. However, I actually got bored and disappointed when I was reading the paper. This is too a large degree caused by the presentation (text and figures) of the material (I agree with all the points by reviewer 1):

We thank the reviewer for these views on the paper.

1) It is not clear what the purpose of the study is. Only in the discussion it's written that the "aim was to use atmospheric observations to benchmark soil carbon models". If this was the aim, the evaluation of just two modules of JSBACH is insufficient (and causes my disappointment). It would be better to clearly state already in the introduction that the aim was to evaluate two soil modules of JSBACH.

We have removed the word "benchmark" from the manuscript, as the aim of this paper is not to do a purely numerical ranking of two different models. We change the title to include 'JSBACH', therefore the scope of this paper should be then visible already in the title.

However, if this is the case, the manuscript would do better as a model evaluation paper in Geoscientific Model Development. Generally, the text is written as a model evaluation study and I don't find important results for the general Biogeosciences. My feeling is that the paper should go beyond a model evaluation and include some more substantial scientific questions, hypotheses and findings in order to fit into Biogeosciences.

The aim of this study was to assess, whether we can use the atmospheric $CO_2$ concentration to evaluate two different soil carbon models and assess, what can we by these means say about the functionality of the soil carbon models. We do address scientific questions in this study. Such as, whether the soil carbon content or environmental responses of the decomposition are more responsible for the differences seen in the results. In writing the revised version of the manuscript we will include scientific questions this study addresses to the introduction.

2) If the "aim was to use atmospheric observations to benchmark soil carbon models" (as stated), I would expect a more detailed description of the assumptions and a detailed analysis on how to use atmospheric CO2 observations for the benchmarking, including how to disentangle the contributions of GPP and Reco on CO2, the role of uncertainties in observations and atmospheric transport, and how different regions contribute to the CO2 seasonality. Especially the later points would help to potentially benchmark soil carbon simulations if different parts of the world.

We will pay more attention to the points raised by the reviewer in the revised version of the manuscript. It is not possible in our current set-up to disentangle the contributions of GPP and Reco, but the GPP is same for both of the model simulations and we re-write the text to remove using the word benchmarking, so that our main aim to see the contribution of two different soil carbon modules in the atmospheric $CO_2$ molar fraction becomes more clear.

We will add text on the observational and atmospheric transport uncertainties. The atmospheric transport model was the same in the both model simulations, so we don't expect this to be causing differences we see in the two different model runs. The role of different regions contributing to the seasonality of the $CO_2$ molar fraction can be studied by setting the land fluxes to zero at certain

regions, as done earlier by Dalmonech and Zaehle (2013). We did not for now consider this to be necessary for this analysis.

3) As already stated by reviewer 1, the text needs substantial rewrite. The text has no clear structure, topic sentences are missing, some chapters are too long (especially 1and 3.1). For example, the first section of the results (3.1) report mainly minor results (including references to the supplement) but does not report the most important results. In addition, I recommend to split this section in further sub-chapters to improve the structure of the text.

Thank you for the advice in improving the re-writing of the manuscript. Concerning the order of the results section: This work was done in a step-wise manner, when first the JSBACH simulations were performed and they were then fed to TM5. Starting with the atmospheric $CO_2$ results (which we would consider being main results of the study) potentially makes it challenging for the reader to follow the story line. Having said that, we will bear this view in mind during re-writing and try to highlight important results before minor results.

4) Figures: I'm sorry, but reading the figures in the main text and in the supplement wasa nightmare! The figures are too small and the grey colours make it almost impossible to distinguish the different model runs and observations. Please improve all figures.

We thank the reviewer for this insight and improve the figures in the revised version of the manuscript.

5) The discussion of GPP is over-simplistic. JSBACH overestimates GPP and has in some regions shifts in the seasonality. Hence it remains unclear which soil carbon model is the better one because the comparison of CO2 seasonality is also affected by wrong simulations of GPP. Could it be an option to force more realistic GPP estimates into JSBACH or mix GPP from data-driven estimates with Rh from JSBACH in TM5?

As this study was also a methodological test for us to see, if using different soil models would indeed result in differences in the atmospheric $CO_2$ cycle, we wanted to only use the 'off-shelf' version of JSBACH. It is obvious that using a full land surface model brings up constraints. What was not successful in writing of the earlier version, was to bear this in mind when interpreting the results. Overall, in this model evaluation exercise, a performance competition of the two different models was not the main goal, but to find out what characteristics of the soil carbon models need improvement. For the YAS model the finding, that the precipitation should not be used as a proxy for soil moisture at this temporal and spatial resolution, is relevant and independent of JSBACHs GPP overestimation.

The reviewer has a suggestion of forcing more realistic GPP into JSBACH. This aim can be achieved by data assimilation, as done by Castro-Morales et al. (2019) where they used another, earlier version of JSBACH. However, this approach would be a study by itself to be done first.

As for using more data-driven GPP estimate, this unfortunately doesn't work in our simulation set-up, as this would break mass-balance of the terrestrial carbon cycle and the litterfall and GPP are strongly coupled in such a model. There would be several biases resulting from this. There certainly is a lot of value in more data-based approaches and that kind of work would be best done by a test-bed set-up similar to works by Wieder et al. (2018). We will now mention this point in the discussion of the revised version.

6) Please describe if permafrost was simulated in the JSBACH runs and how the simulation or non-simulation of permafrost contributed to soil carbon simulations.

Permafrost was not included in the simulations and we will mention this in the new version of the manuscript clearly. Recent model development adding permafrost to JSBACH includes only the YAS model (Castro-Morales et al., 2018). Including permafrost would increase the carbon stocks in high latitude regions. The exact influence on the atmospheric signal is speculation without the actual model runs, but it would be expected that the seasonal cycle of heterotrophic respiration in high latitude regions would be dampened, since the active layer producing heterotrophic respiration would be thinner.

Figure 1: Explain the numbers in the caption.

Thank you for noticing this was missing. We have now changed the caption to be:

*Locations of GAW stations, denoted as black dots, and different TransCom regions (different numbers denote the different TransCom regions in this study) as different colors.*

Figure 2: Even if the soil carbon stocks have been already evaluated, it would be still helpful to add 1 or 2 maps from an observation-based product for comparison.

Sure, we will do this.

Table 2:  There seems to be a mistake in the results for TER, as those can't be the same numbers.

Yes, thank you, the YAS number should be 155 PgC.

**References**

Castro-Morales, K., Kleinen, T., Kaiser, S., Zaehle, S., Kittler, F., Kwon, M. J., Beer, C., and Göckede, M.: Year-round simulated methane emissions from a permafrost ecosystem in Northeast Siberia, Biogeosciences, 15, 2691–2722, https://doi.org/10.5194/bg-15-2691-2018, 2018.

Castro-Morales, K., Schürmann, G., Köstler, C., Rödenbeck, C., Heimann, M., and Zaehle, S.: Three decades of simulated global terrestrial carbon fluxes from a data assimilation system confronted with different periods of observations, Biogeosciences, 16, 3009–3032, https://doi.org/10.5194/bg-16-3009-2019, 2019.

Dalmonech, D. and Zaehle, S.: Towards a more objective evaluation of modelled land-carbon trends using atmospheric $CO_2$ and satellite-based vegetation activity observations, Biogeosciences, 10, 4189–4210, https://doi.org/10.5194/bg-10-4189-2013, 2013.

---

## Author Response (AR1)

Reply to reviewer #1 by Thum et al. (The original comments by the reviewer are in violet and the replies by the authors are in black.)

Thum and colleagues present an interesting study looking at how different soil models influence atmospheric CO2 fluxes, which can be compared with ground and space-based observations. The study is thorough and well written, but in my estimation,it somewhat glosses over some of the caveats that should likely be considered with trying to evaluate component fluxes (here HR), with atmospheric CO2 observations that serve as a proxy for NEE.

We thank the reviewer for these in-depth comments and views. We hope we are able to address the concerns the reviewer is bringing up in our replies and in the revised version of the manuscript in a satisfactory manner.

Major concerns: Introduction, I found the basic primer on the global carbon cycle a little pedantic at times, with phrases like "Photosynthesis takes place in the green plant parts". As well as wandering, combining information on soil respiration and soil C stocks without really developing these ideas in a focused way that supports the direction or intent of the paper. I realize this is more stylistic that substantiative comment, but would encourage the authors to revise the introduction with focused topic sentences for each paragraph that introduces background literature and develop ideas in a way that focuses attention on the work to be presented.

We thank the reviewer for this advice in improving the introduction. We will re-write the introduction to be less pedantic and also take on the other good hints forward provided here. The reason we wrote the sentence "*Photosynthesis takes place in the green plant parts*" this way was to highlight the ending of this sentence: that the remote sensing is therefore able to detect the terrestrial photosynthesis.

I'm trying to wrap my head around how biases in the magnitude (and timing) of GPP simulated by JSBACH confound results presented here

In the JSBACH simulations the biosphere has been first spun-up to steady state in 1860 and the current land sink is resulting from the simulation period between 1860 to present. Therefore in 1860 the global net land $CO_2$ balance is zero, and if the gross primary production is then overestimated, the autotrophic and heterotrophic respiration are then also overestimated. See below for more discussion on the mean and seasonality biases and their relevance for comparing our two model formulations.

and I fundamentally disagree with the statement on Line 392.

The sentence in line 392 is: "*The biases between XCO$_2$ from satellite retrievals and the model results originating from the JSBACH simulations are relatively large and this is likely caused by the use of a posteriori ocean fluxes from the CTE2016.*"

The biases we talk about in the sentence in line 392 is connected to the absolute differences in the $CO_2$ concentration values that were reported in lines 294 and 322 [we should have mentioned also the site level observations, and not only the satellite observations in this sentence].

This bias we mean here is not connected to the sink or source terms separately but to the net land sink that we report in Table 2 to be -1.68 PgC for CBA and -1.75 for YAS, and how it differs from the net land sink from the CTE2016 framework. With Fig. S10 (Fig. S13 in the revised version) we aimed to demonstrate how development of the bias in the absolute $CO_2$ values is in line with these

different net land sink estimates, as we have explained in lines 394-397.

Re-writing the sentence in line 392 requires emphasizing better which bias we are meaning. We agree with the reviewer that within the JSBACH model the modelled GPP is overestimated (against FLUXCOM, but more in line with the estimations from Welp et al. (2010), as mentioned in the manuscript) and the timing doesn't match perfectly. We will bring this up more in the new version of the manuscript, adding that also the bias of GPP influences the modelling skill of the system and not only the performance of the soil carbon model itself (see below for the places and excerpts stating this).

The new version of the sentence that was previously in line 392 is now (located in lines 476-477):

*"The differences in absolute $CO_2$ and $XCO_2$ levels against the surface observations and the satellite retrievals, respectively, with modelled $CO_2$ are caused by the modelling system, but this bias does not influence the analysis performed."*

Should SI4 and Fig 2 be combined into one display item in the main text, using the same y-axis units for both?

The Fig. 2 displays a soil carbon map and Fig. S4 the anomalies from the turnover rates. If we put the color code to be the same for the both models, the spatial patterns won't be anymore visible. Since the reviewer 2 requested to also show observation based soil carbon maps, we will add observations to Fig. 2 and still keep Fig. S4 and Fig. 2 separate.

In the revised version of the manuscript the Fig. 2 is now Fig. S11 (including soil maps also from data based estimates from SoilGrids and WHSD) and the previous Fig. S4 is now Fig. 4.

Could you show the annual cycle of NPP or HR globally, in addition to the latitudinal bins shows in SI2? [ maybe this goes into SI ].
Would similar plots of regional & global AR or NPP also be helpful (in SI)?

We now show a plot of NPP, heterotrophic respiration (HR) and NEE, both global cycles (Fig. 2) as well as divided into latitudinal bins (Fig. 3). Additionally we have added a plot of autotrophic respiration (AR) to the supplement (Fig. S2), showing both the global and latitudinally divided yearly cycles.

It's not really clear that the magnitude seasonal cycle of HR with YAS is so large in mid latitudes (30-60N; Fig 4), and is of equal and nearly opposite magnitude to the high GPP biases in this region, resulting in a lower than observed NEE (Fig 6-7).

Magnitude of the seasonal cycle at these latitudes by YAS is 2.5 Pg month$^{-1}$, and the GPP bias in this region is approximately 0.05 PgC day$^{-1}$ = 1.5 PgC month$^{-1}$. How the difference between the heterotrophic respiration from the YAS and CBA models influence the global and latitudinally segregated NEE values is shown in the figures 2 and 3 in the revised manuscript.

I'm wondering how any version of JSBACH captures appropriate seasonal peak to trough dynamics of NEE Fig 6 given biases in the magnitude of GPP fluxes? This means that the timing and or magnitude of AR or HR fluxes must be compensating to generate reasonable seasonal cycle of NEE.

The annual magnitudes of the autotrophic and heterotrophic respiration have been given in Table 2 in the first and revised versions. The autotrophic respiration is relatively large. In the revised

version of the manuscript we will show the annual cycles of NPP and NEE globally (Fig. 2) and in latitudinal bins (Fig. 3) and autotrophic respiration similarly in the supplementary material (Fig. S2). The compensation by the autotrophic and heterotrophic respirations become visible in these plots and we also discuss these plots in the revised manuscript.

Given known biases in the seasonal cycles of GPP (Fig S2), what modifications are needed to improve the representation of plant and soil dynamics in JSBACH?

In the data assimilation study by Castro-Morales et al. (2019), where satellite observed FAPAR and atmospheric $CO_2$ molar fractions were used in the assimilation, the maximum LAI value in the tropics was lowered via optimization. This itself will not enhance the seasonal cycle in the tropics, but would bring its absolute value down. To improve the seasonality of the tropical GPP, re-parametrization of the phenology model for the tropical plant functional type would likely help.

The timing is not so much off in the region 30ºN - 60ºN, but for the phenology during the senescence period there might be room for improvement. North of 60º the modelled GPP is peaking too late, but the start of the growing season is occurring at the same time with the FLUXCOM results.

A study done for Finland, comparing growing season onset by satellite observations to JSBACH simulation output found out that it was enough to improve the onset of the deciduous forest by re-parameterizing the phenology module, however, to improve the onset of the coniferous forest it was necessary to add seasonality to the temperature responses of the photosynthesis parameters (Böttcher et al., 2016). It is therefore not so straightforward to say, which changes would improve the seasonal cycle of GPP. Improving the phenology cycle might be a step forward, but in the tropics how the plants experience dry conditions might be also off. The nutrient cycles were not modelled in our study. It is likely that including the nitrogen cycle the GPP values would be lower, as some vegetation would be nitrogen limited.

For the soil dynamics, based on the results of this study, we strongly recommend moving towards using the soil moisture as a driver for the YAS model. This was already mentioned in the earlier manuscript version.

It is not straightforward to come up with a solution for fixing the GPP, as that would be a study on its own and without thorough analysis and further model development (being out of the scope for this paper) it is not obvious which solutions would be required. We have now added to the lines 426-431 the following text:

"*Furthermore, the high GPP values predicted in the current run would likely be lower, if the nutrient cycles of nitrogen and phosphorus were included in the used version of JSBACH. Beside using a JSBACH version with nutrient cycles, further development work in the phenological cycle could improve the estimated GPP. The difference of the modelled GPP to the FLUXCOM product (Fig. S9) suggests that the maximum leaf area index might be overestimated in the tropics. Also, the timing of the phenological cycle north of 60 N might benefit from re-parametrization.*"

In summary,could one erroneously chose an inferior soil biogeochemical model that give 'better' NEE fluxes with atmospheric observations, but that's fundamentally just masking over /compensating biases in GPP? This is, 'getting the right answer for the wrong reasons?

The reviewer is right that the biased GPP might influence the results. Therefore we will modify the wordings of this manuscript so that we will not talk about benchmarking, but will emphasize that the aim was to see how well we can try to assess the differences between the soil carbon

formulations within this kind of system. We now acknowledge that when it comes to ranking the models, we would need a more data-driven system as testbed, where the soil carbon models can be forced with certain plant productivity inputs (Wieder et al., 2018) or a systematic assessment of several variables against observations. The aim of this study was a comparison of the two soil carbon models, which both had the same GPP input, and while the GPP bias compared to observations does have implications for numerical benchmarking, it does not take away the importance and conclusions of this work.

While the GPP of the JSBACH is not a perfect match to observations, it is the same in both model simulations and the aim was to evaluate the soil carbon modules within a global land surface model that includes several different process descriptions, e.g. also fires and land use changes. JSBACH is also part of the Earth System Model of the Max Planck Institute and an IPCC model, and it is state-of-the-art land surface model.

To make this point raised by the reviewer in the manuscript, we have written in lines 423-426, when discussing the JSBACH results:

"*That GPP of JSBACH is biased high compared to observations is likely of secondary importance to our study comparing two model formulations, because GPP was the same for both formulations and the GPP bias did not lead to strong biases in the seasonal cycle predictions in different latitudinal zones were (Fig. S9). However, to assess the absolute skill of each model formulation in terms of net ecosystem exchange, GPP biases need to be reduced.*"

And to lines 472-475, when discussing the atmospheric $CO_2$ results:

"*The simulated GPP had a larger magnitude and some bias in its seasonal cycle, and therefore its evaluation against atmospheric $CO_2$ observations is influenced by it. Even though the atmospheric observations provide a valuable and informative comparison for the model results, their use as a benchmark metric needs careful consideration.*"

and have added to the Conclusions in lines 513-514 the following:

"*Also, the evaluation was done within a land surface model that is biased in its GPP predictions when compared to an upscaled GPP product and this hampers the use of atmospheric $CO_2$ as a numeric benchmark.*"

Throughout on display items, the shades of grey make it kind of hard to distinguish models and observations. This is especially true for Figs 5-7). Is there any harm in using colors?

We have added colors to all of the figures that were black and white in the earlier version.

Minor and technical concerns: There are enough abbreviations in the text that it some what distracts from the readability of the manuscript. I'd encourage removing some of these less standard ones if possible (e.g. SCA, GAW).There are also several very short paragraphs (even just one sentence long), that should either be further developed or merged into related paragraphs.

We have removed the abbreviations for SCA (with some exceptions in the figures to save space) and GAW. We have now paid attention to the length of the paragraphs and have developed short ones or merged them with other paragraphs, as suggested.

L19: the IAV of fluxes are never communicated here (although they could be easily brought into text and display items.

We have added the annual values to the plots that show seasonal cycles of NPP, TER and NEE.

L26: Maybe remove 'natural'

Removed.

L39: I might include van Gestel et al's 2018 critique of the Crowther paper to make this point.

Thank you for bringing up this point, we have now added a sentence (lines 27-29):

"*For example, while Crowther et al. (2016) concluded in data-based analysis that large carbon stocks will lose more carbon due to warming conditions, but van Gestel et al. (2018) questioned this view by an analysis based on more comprehensive dataset.* "

L56: The connection between benchmarking global soil C models (the topic of the last paragraph) and global CO2 flux observations is a little rocky and unclear. Reading between the lines, I think it's a very good idea- but the connections about how / why it may be considered a useful way to evaluate soil biogeochemical models should be clarified. That said, others have recently used a similar approach (Basile et al 2020), which could be useful in contextualizing the introduction of the present work.

Thanks, we have now used the Basile paper in bringing this work better to the context. We have now improved in the introduction section about the connection of atmospheric $CO_2$ molar fractions and soil model evaluation.

I'm also assuming the authors will discuss some of the assumptions being made in evaluating HR fluxes with atmospheric CO2 observations that may include biases in the timing and magnitude of GPP and AR fluxes from (JSBACH), potential errors imparted by the atmospheric transport model (TM5), or challenges in interpreting total column CO2 observations- especially from space. Maybe it's worth foreshadowing some of these in the introduction?

We thank the reviewer for this insight, and find the idea of bringing them up in the introduction a good idea. In the new introduction we have now introduced these points and they are also mentioned in the discussion.

See also refs from Keppel-Aleks et al (below).

We also thank the reviewer for pointing to these useful references that we have used in the revised version of the manuscript. We used the paper by Keppel-Aleks et al. (2011) to highlight the influence of several factors to the column $XCO_2$ profiles. We referenced to Keppel-Aleks et al. (2013) as a work that has used column XCO2 profiles in model evaluation.

Line 67: this sentence seems awkwardly phrased, maybe drop 'in' and 'far'.

We'll rephrase this sentence.

Line 84: Is there reference for TM5, or example of where / how it's been used?

In the earlier version we had added more information about TM5 only in section 2.3. We have written in the revised version already to the introduction the following sentence (line 55):

"*In this work we used a three-dimensional atmospheric chemistry transport model TM5 (Krol et al., 2005, Huijnen et al.,2010.*"

Line 97: Randerson et al 1997 found that the timing of litterfall was important for controlling the timing of HR fluxes. Is the same true in JSBACH? How well does the model simulate this phenology?

As shown in Table 3, the decomposition is for a large part regulated by the environmental conditions. For completeness, we now checked the correlations between litterfall and heterotrophic respiration (HR)  similarly as we did for the environmental drivers in Table 3: There is a positive significant correlation between the litterfall and heterotrophic respiration only in region 30 N - 60 N with both of the model versions. The increase of heterotrophic respiration is anyhow preceding the litter flux. We have added a plot of this to the supplement (Fig. S8) and discuss this in the manuscript.

An interesting point in the study by Randerson et al is that changing litter quality would be making changes to the decomposition they are seeing in their study. This is something that we could actually test with the YAS model and we have now mentioned this and the previous point in the Discussions section (lines 403-407) in the revised version of the manuscript as:

"*The observations show that the litterfall has strong influences on heterotrophic respiration (Chemidlin Prevost-Boure et al., 2010), but this process is not included in the used models, so at seasonal timescales in the different latitudinal zones no clear influence of litterfall driving $R_h$ was seen. However, changes in the chemical composition of litterfall is considered to be potential reason for changes in the amplitude of atmospheric $CO_2$ (Randerson et al., 1997) and this is something we could study with the YAS model.*"

Line 140, I realize it's likely in your previous publications, but is it worth noting how litter chemistry from JSBACH PFTs is translated onto the YASSO litter quality definitions? The way this is presented in kind of confusing & disconnected

We have clarified this in the revision of the manuscript. The division of the incoming litter from the JSBACH model per PFT is based on observations from different ecosystems (Trofymow et al., 1998; Berg et al., 1991a, 1991b; Gholz et al., 2000). We have added this information to the new version of the manuscript.

Line 143 define PFT

Thank you, we provide a definition for it here.

Line 159 constraints for what?

This comment refers to the sentence: '*The 3-hourly meteorological fields from ECMWF ERA-Interim (Dee et al., 2011) were used as constraints.*' We have modified this sentence to say that the TM5 model is run with the 3-hourly meteorological data from ERA-Interim (lines 158-159):

"*The 3-hourly meteorological fields from ECMWF ERA-Interim (Dee et al., 2011) were used as forcing to run the TM5 model.*"

Line 175 is 'atmospheric' redundant here?

Thanks, removed.

Line 205: Should this be S2.

Thanks, corrected.

Line 250: Why run statics on YAS fluxes and alpha, when the model uses precipitation to moderate decomposition rates (eq5)?

As explained in the manuscript in line 378, the precipitation is used as a driver in YAS, since it has been considered to be a proxy for soil moisture. Precipitation has the advantage that it is much easier to observe than soil moisture and was thus the applied as driving variable in the development of the heavily observation-based YASSO. While an approximation of soil moisture effects by precipitation works relatively well at the annual timescales, in which this model has been originally developed, this comparison here in Table 3 shows that it is not justified at monthly scale. We have commented this in the earlier manuscript version in line lines 384-386 (new version lines xx) with:

"*Precipitation has been originally used in the YAS model as a proxy for soil moisture, since enough accurate soil moisture observations for model development haven't been available. Clearly, this idea needs reconsideration as our results show that at zonal spatial scales and monthly temporal scale the Rh from YAS is not at all correlated to soil moisture variable α.*" [New version, lines 415-418: "*Precipitation was originally used in the YAS model as a proxy for soil moisture, since enough accurate soil moisture observations for model development were not available. Clearly, this idea needs reconsideration as our results show that at zonal spatial scales and monthly temporal scale, Rh from YAS is not at all correlated to the soil moisture.*"]

and in the Discussion and in line 423-424 (new version line 520) in the Conclusions: "*This suggests that use of precipitation as a proxy for soil moisture might not be sensible in sub-annual time scales.*"

Are the correlations between environmental drivers and HR fluxes really that surprising or interesting (Table 3, Line 241-254 & 366)? The models have these assumptions a priori (Methods, Fig. S1). In places with large seasonal variation in soil or air temperature (arctic), temperature is important control over seasonal HR fluxes. In places with little annual variation in temperature (tropics) moisture is a more important control. I'm not really sure what readers are supposed to learn from this analysis?

While we do have these clear response functions that are driving the soil carbon models, it is not necessarily clear at which part of the response we're in these different ecosystems. E.g. in the tropical zone, area 10ºS - 0º in Fig. S5, the YAS model is actually reaching the saturation level in respect to moisture.

And as the reviewer later mentions, the litter flux could play a role, but this is not a role in our model and these high correlation values to the environmental drivers already suggest that. These correlations also function as comparison for the soil moisture vs. YAS $R_h$, as mentioned.

Fig 6: Dotter is not a word.

Thanks, it was a typo. In the revised version we will have colors in this figure and not have a dotted line there.

Fig 6 & 7, can uncertainty estimate (or interannual variability) be shown for observations?

Thanks for the suggestion, we have now added interannual variability of the seasonal cycle amplitude to the plots. Additionally we have calculated an estimate for the uncertainty in the observations so that we calculated the difference between the fit to the observations and the actual observations and estimated the standard deviations from these values for each day with the observations. This is now represented in the figures as shaded area.

*Table 3, why not just report r values, so negative correlations can be more clearly illustrated. Can't statistically significant correlations be highlighted (not just results withhigh r2)?*

Thanks for this suggestion, we have now done so in the revised version.

*L345, I'm not sure better agreement with CMIP5 models is necessarily a good thing,based on Kathe's work. Moreover, the calculation of global turnover times seems to mask important regional patterns. Instead, see Koven et al. 2017.*

We considered adding this metric here, since it has been used, but this was more to complement the Figure S4 that was showing a map of the turnover rate anomalies by the two models. We agree with the reviewer that a global value does not include many important features and add here also a plot similar to Koven et al. 2017 (now Fig. 5), where the turnover rates are shown as a function of air temperature and the precipitation is visible via coloring. We also removed the sentence from the conclusions that was re-stating the sentence, so that it has now less emphasis.

*L395: where are results supporting these claims shown?*

The sentence in this line is: '*The global land sink of JSBACH is approximately -1.7 PgCyr −1 (Table 2) and therefore the used ocean fluxes cause a bias to the simulated atmospheric $CO_2$ molar fraction.*'

We used posteriori ocean fluxes from the CTE2016 in this study. These ocean fluxes have been optimized using the terrestrial carbon cycle of the CTE2016, the SiBCASA-GFED4 model (van der Velde et al., 2014), and the fire emission fluxes that have been estimated from satellite observed burned area (Giglio et al., 2013). The fossil fuel emissions have not been optimized.

Since the optimization has been done with this other set of terrestrial carbon fluxes, this would be the likely cause for the bias (see also above). The fire fluxes of the CTE2016 are also different to JSBACH, so that could be another source for the discrepancy. We have modified this paragraph to add this point and also clarified why we consider the ocean fluxes to be causing this bias.

*L405: This doesn't seem like a standalone paragraph, nor is it really clear how it relates to the results being discussed regarding carbon tracker and JSBACH land C uptake.*

Thanks for noting this, we have added the information content to a paragraph that discusses the results from GOSAT and flux observations.

*L408: what biases are not important here?*

We admit that this line is unclear. We were referring here to the biases in the absolute $CO_2$ molar fraction values, and that these are not important, since the differences in the absolute values are not playing a role, since we concentrated in this analysis on the seasonal cycles and only mentioned the bias in the absolute $CO_2$ for completeness. We have rephrased this sentence to be:

*"The differences in absolute $CO_2$ and $XCO_2$ levels against the surface observations and the satellite retrievals, respectively, with modelled $CO_2$ are caused by the modelling system, but this bias does not influence the analysis performed."*

L414: while I agree with this statement, however it's not done in the work presented. Multiple benchmarks, however, could be presented, again see Koven et al. 2017, or Todd-Brown's work that's already cited). Indeed it seems for a flux based analysis like this some more rigorous evaluation of upstream fluxes (e.g. GPP) is pretty important.

The sentence on the line is: "*We demonstrated how atmospheric $CO_2$ observations can be used to benchmark soil carbon models and that it is important to benchmark models across several different variables.*" We will remove the word benchmarking from the revised version of the manuscript, since we did not perform rigorous numerical benchmarking in this study, but were evaluating two different soil carbon models within one global land surface model within the constraints given by that system.

We have removed the word "benchmark" from the manuscript, as the aim of this paper is not to do a purely numerical ranking of two different models. We change the title to include 'JSBACH', therefore the scope of this paper should be then visible already in the title.

However, if this is the case,the manuscript would do better as a model evaluation paper in Geoscientific Model Development. Generally, the text is written as a model evaluation study and I don't find important results for the general Biogeosciences. My feeling is that the paper should go beyond a model evaluation and include some more substantial scientific questions, hypotheses and findings in order to fit into Biogeosciences.

The aim of this study was to provide a proof of concept that atmospheric $CO_2$ concentrations are a useful variable to evaluate soil carbon models and to apply this to two different model formulations of JSBACH to reveal what we can learn about the functionality of the soil carbon models. Our aim is not a rigorous numerical benchmarking, but to see what causes differences we see in the model results versus the observations.

Furthermore, we do address scientific questions in this study, such as whether the soil carbon content or environmental responses of the decomposition are more responsible for the differences seen in the results. We therefore see our manuscript well suited for Biogeosciences. In writing the revised version of the manuscript we will include the following scientific questions this study addresses to the introduction:

*How do can we use a land surface model together with a transport model to evaluate soil carbon model and what problems do we face when doing that?*
*What is the role of soil carbon stocks, the variables driving their decomposition and the functional dependencies of those variables on modelled heterotrophic respiration at global scale and how this leads to differences in the atmospheric CO$_2$ signal?*

In the revised version of the manuscript we now better highlight how the changes in the heterotrophic respiration influence the modelled terrestrial NEE, making it more visible how the process of changes in heterotrophic respiration transfers to the atmospheric signal. This therefore better illustrates the process chain and connects better the terrestrial modelled carbon cycle to the atmospheric $CO_2$ signal. Since the representation of the land carbon cycle and its gross fluxes are similar across most state-of-the-art DGVMs, such analysis provides more universal insight than just

a JSBACH evaluation. Also we show environmental dependencies of the modelled turnover rates, which brings them better into context of ecological understanding. Additionally we show a latitudinal comparison of the modelled soil carbon stocks to the observation-based estimates. We believe that all these new contributions (suggested by the reviewers) tie this manuscript better in line with the Biogeosciences.

2) If the "aim was to use atmospheric observations to benchmark soil carbon models" (as stated),  I would expect a more detailed description of the assumptions and a detailed analysis on how to use atmospheric CO2 observations for the benchmarking, including how to disentangle the contributions of GPP and Reco on CO2, the role of uncertainties in observations and atmospheric transport, and how different regions contribute to the CO2 seasonality.  Especially the later points would help to potentially benchmark soil carbon simulations if different parts of the world.

We have paid more attention to the points raised by the reviewer in the revised version of the manuscript. It is not possible in our current set-up to disentangle the contributions of GPP and Reco, but the GPP is same for both of the model simulations and we re-wrote the text to remove using the word benchmarking, so that our main aim to see the contribution of two different soil carbon modules in the atmospheric $CO_2$ molar fraction becomes more clear.

We have added text on the observational uncertainty concerning the surface $CO_2$ observations (in Section 2.4), spaceborn $XCO_2$ observations and atmospheric transport uncertainties (in Introduction)**.** The atmospheric transport model was the same in the both model simulations, so we don't expect this to be causing differences we see in the two different model runs. For the modelled atmospheric signal we also have contributions from the ocean fluxes and anthropogenic emissions. Errors in these estimates will also influence the atmospheric $CO_2$, but they were the same for both model simulations. The role of different regions contributing to the seasonality of the $CO_2$ molar fraction can be studied by setting the land fluxes to zero at certain regions, as done earlier by Dalmonech and Zaehle (2013). We did not for now consider this to be necessary for this analysis.

3) As already stated by reviewer 1, the text needs substantial rewrite. The text has no clear structure, topic sentences are missing, some chapters are too long (especially 1 and 3.1).  For example, the first section of the results (3.1) report mainly minor results (including references to the supplement) but does not report the most important results. In addition,  I recommend to split this section in further sub-chapters to improve the structure of the text.

Thank you for the advice in improving the re-writing of the manuscript. Concerning the order of the results section: This work was done in a step-wise manner, when first the JSBACH simulations were performed and they were then fed to TM5. Starting with the atmospheric $CO_2$ results (which we would consider being main results of the study) potentially makes it challenging for the reader to follow the storyline. Having said that, we bore this view in mind during re-writing and tried to highlight important results before minor results.

In order to achieve this, we put the flux results before the stock comparisons in section 3.1. Further, we have divided 3.1. into further sub-chapters (3.1.1. Flux comparisons, 3.1.2. Stock comparisons, 3.1.3. Box model) as suggested. The reviewer also noted that the introduction has been too long. We have tried to shorten it in the revised version. Unfortunately the section 3 that was also too long according to the reviewer is now longer, due to the suggestions from the reviewers, but we hope that using the subsections for this section makes it easier to follow. Effort has been made to add missing topic sentences.

4) Figures: I'm sorry, but reading the figures in the main text and in the supplement wasa nightmare! The figures are too small and the grey colours make it almost impossible to distinguish

the different model runs and observations. Please improve all figures.

We thank the reviewer for this insight and improve the figures in the revised version of the manuscript. We apologize the inconvenience -- the size of the figures was caused by following the template provided by the publisher and not noticing that it should have been wider for two column figures. We have now made the figures bigger.

5) The discussion of GPP is over-simplistic. JSBACH overestimates GPP and has in some regions shifts in the seasonality. Hence it remains unclear which soil carbon model is the better one because the comparison of CO2 seasonality is also affected by wrong simulations of GPP. Could it be an option to force more realistic GPP estimates into JSBACH or mix GPP from data-driven estimates with Rh from JSBACH in TM5?

As this study was also a methodological test to see if using different soil models would indeed result in differences in the atmospheric $CO_2$ cycle, we wanted to only use the 'off-shelf' version of JSBACH. This is also in line with our goal of a proof of concept study for evaluation of other land surface models, which would likewise likely not be optimized in their performance. It is obvious that using a full land surface model brings up constraints. What was not successful in writing of the earlier version, was to bear this in mind when interpreting the results. Overall, in this model evaluation exercise, a performance competition of the two different models was not the main goal, but to find out what characteristics of the soil carbon models need improvement and how their different structures influence the results and how the differences transfer all the way to the atmospheric $CO_2$ molar fractions. For the YAS model the finding that the precipitation should not be used as a proxy for soil moisture at this temporal and spatial resolution, is relevant and independent of JSBACHs GPP overestimation.

The reviewer has a suggestion of forcing more realistic GPP into JSBACH. This aim can be achieved by data assimilation, as done by Nyawira et al. (2016) and Castro-Morales et al. (2019) where they used other, earlier versions of JSBACH. However, this approach would be a study by itself to be done first.

As for using more data-driven GPP estimate, this unfortunately doesn't work in our simulation set-up, as this would break mass-balance of the terrestrial carbon cycle and the litterfall and GPP are strongly coupled in such a model. There would be several biases resulting from this. There certainly is a lot of value in more data-based approaches and that kind of work would be best done by a test-bed set-up similar to works by Wieder et al. (2018).

We have now added a point to the potential biases in the atmospheric $CO_2$ signal caused by the modelled GPP to two places in the discussion and once in the conclusions (see the responses to reviewer 1).

6) Please describe if permafrost was simulated in the JSBACH runs and how the simulation or non-simulation of permafrost contributed to soil carbon simulations.

Permafrost was not included in the simulations and we will mention this in the new version of the manuscript clearly. Recent model development adding permafrost to JSBACH includes only the YAS model (Castro-Morales et al., 2018). Including permafrost would increase the carbon stocks in high latitude regions. The exact influence on the atmospheric signal is speculation without the actual model runs, but it would be expected that the seasonal cycle of heterotrophic respiration in high latitude regions would be dampened, since the active layer producing heterotrophic respiration would be thinner.

We have now added to lines 447-452 the following text:

*"This JSBACH version also didn't have permafrost described. If permafrost would be modelled, the seasonal cycle of heterotrophic respiration at high latitudes would likely be dampened, as the depth of the active layer determines the amount of soil capable of respiring. The YAS model has been used in a JSBACH version containing permafrost in a study concentrating on the Russian Far East (Castro-Morales et al., 2018). Both, CBA and YAS, were originally developed for mineral soils and for applications with organic soil, so model development and testing at smaller than global scale could be useful."*

Figure 1: Explain the numbers in the caption.

Thank you for noticing this was missing. We have now changed the caption to be:

*"Locations of GAW stations, denoted as black dots, and different TransCom regions (different numbers denote the different TransCom regions in this study) as different colors."*

Figure 2: Even if the soil carbon stocks have been already evaluated, it would be still helpful to add 1 or 2 maps from an observation-based product for comparison.

We have added maps from SoilGrids and HWSD (Fig. S11) products for the comparison with the soil carbon stocks, as suggested. Additionally, we added a figure showing a latitudinal gradient of the carbon stock, to better illustrate the differences between the modelled and observation-based estimates (Fig. S12).

Table 2: There seems to be a mistake in the results for TER, as those can't be the same numbers.

Yes, thank you, the YAS number should be 155 PgC.

The results at surface sites show that CBA largely overestimated  seasonal cycle amplitude at high northern latitudes, whereas YAS almost consistently underestimated the  seasonal cycle amplitude in the Northern Hemisphere. CBA captured the seasonal cycle patterns better than YAS across different latitudes. Overall, the YAS model showed biases in the

atmospheric $CO_2$ cycle at temperate latitudes in the Northern Hemisphere, whereas the CBA model had biases in the high latitudes in the Northern Hemisphere.

**3.3 Column $XCO_2$ comparisons for TransCom regions**

This evaluation of the two soil modules against satellite column $XCO_2$ was carried out for the different TransCom (TC) regions (Fig. **??**). The comparison was based on seasonal cycle amplitudes and $r^2$ values similar to the surface site evaluation. Not all the TC regions show a clear seasonal cycle, such as regions in South America (TC regions 3 and 4), northern part of Africa (TC=5) and Australia (TC=10). For completeness we show the analysis also for these regions in Table S5. For regions with clear seasonal  cycles we used the ccgcrv curve fitting procedure available from NOAA (https://www.esrl.noaa.gov/gmd/ccgg/mbl/crvfit/crvfit.html, (**?**)), but for regions with missing data or no clear seasonal cycle, we averaged over all years of data.

To further illustrate the results from this comparison, we show data for two regions having a clear seasonal cycle. In TC region 2, the southern part of North America, CBA is more successful in capturing the observed  seasonal cycle amplitude than YAS (Fig. **??**a), even though CBA reaches the minimum $XCO_2$ later than observations. YAS underestimates  the seasonal cycle amplitude by 56% and has a different seasonal pattern than observations, so the minimum is reached earlier than in the observations and also the shape during the summer period  differs from the observations. In Europe, TC region 11,  both models capture the  seasonal cycle amplitude (Fig. **??**c, Table S5) and the seasonal cycle in the first part of the year. The increase of $CO_2$ is not as well captured by the simulations. The time series of seasonal cycle amplitudes predicted by the CBA and YAS models(**??**c, d) do not correlate significantly with the observations.

[revised manuscript text omitted]
  estimated of 146 ($\pm$ 19)  $PgCyr^{-1}$ (for 1980-1999) (**?**) and estimates based on isotope observations  are 150 to 175 $PgCyr^{-1}$  (for 1980-2009) (**?**). That GPP of JSBACH is  biased high compared to observations is likely of secondary importance to our study comparing two model formulations, because GPP was the same for both  formulations and the GPP bias did not lead to strong biases in the seasonal cycle predictions in different latitudinal zones were (Fig. S9). However, to assess the absolute skill of each model formulation in terms of net ecosystem exchange, GPP biases need to be reduced. Furthermore, the high GPP values predicted in the current run 
[revised manuscript text omitted]

---

## Author Response (AR2)

**Reply to reviewer 1, 9.9.2020**

(The reviewer comments are in violet and the replies from the authors are in black).

*I appreciate the efforts made to revise this paper. Two outstanding questions from the discussion remained and I have a number of suggestions re. display items. None of these are large enough to warrant another review of the paper once addressed.*

We thank the reviewer for going through the manuscript thoroughly again and the new feedback to improve the manuscript.

*Line 420-430 I don't really understand why (or agree with) the assertion that potential biases in GPP (Fig S9) don't influence in evaluation of the soil models and NEE fluxes that are simulated by the model? Yes, they're consistently high between YAS and CBA, but if you're ultimately trying to evaluate with observations based on the net fluxes (or atmospheric CO2 concentrations) then biases in the gross fluxes being simulated need to be considered? As opposed to brushing this concern aside, why not discuss this limitation in the approach?*

This was already a concern for the reviewer in the first review ground and we addressed this by adding text about this issue to three different places. However, since it is still required to have more emphasis, we have added text to the Discussion (lines 424-430):

"*Fig. S9 shows that the bias relative to FLUXCOM exists throughout most of the Northern Hemisphere and the tropics, but has only minor influence on the seasonal cycle of GPP. The high estimate of GPP will propagate into larger NPP, litter input and therefore also simulated heterotrophic respiration and soil carbon stocks. While this may contribute to a slightly larger simulated seasonal cycle of atmospheric $CO_2$ at northern stations, it is unlikely that this will affect our conclusions on the impact of the different soil formulations on the ability of JSBACH to simulate the seasonal cycle of heterotrophic respiration and the residence time of carbon in soil, and as a consequence, its ability to reproduce observed seasonal cycle of atmospheric $CO_2$ or its longterm trend.*"

and Conclusions (lines 540-551):

"*The evaluation was done within a land surface model that overestimates GPP in comparison to an upscaled GPP product and this hampers doing benchmarking using this modeling system. Since the model is run to a steady-state during the spin-up procedure, it also leads to other biases in the modelling system (influencing e.g. autotrophic respiration). Overestimated GPP leads to an enhanced litter input to the soil. This causes comparing the magnitudes of the soil carbon pools to the actual observations cumbersome, as the overestimated litter fall causes biases in the model estimates. In this study the magnitudes of simulated soil carbon are therefore not as good as the spatial patterns as an indicator for the model performance (such as latitudinal gradient). The other downside of the GPP biases is their influence on the estimated NEE. Due to the biases in the timing and magnitude of the other carbon fluxes, it is challenging to use $CO_2$ as a benchmark to heterotrophic respiration. However, in our study the two soil models lead to pronounced differences in the atmospheric $CO_2$ and we were also able to locate latitudinal regions, where the models had most issues. Therefore, this approach provides a method to evaluate how the changes in the heterotrophic fluxes further influence the atmospheric signal and helps to track which geographical areas are contributing to the questionable*

*model performance.*"

This concern extends to the discussion of soil C pools and turnover times. With large positive biases in GPP and small soil C pools the mean turnover times in YAS are really small (14 years). To me this suggests the turnover times and fluxes simulated by this model kind of crazy, a sentiment that seems confirmed by results from Fig 7 & 8 where YAS misses the seasonal cycle in CO2 measurements at multiple scales. If the paper's intent is "Evaluating two soil carbon models", should conclusions about the models in question be more strongly worded? For example, although "The YAS model better captured the magnitude and spatial distribution of soil carbon stocks globally", these stocks likely wouldn't look so good if the model received lower inputs (NPP > 75 Pg C/y!). You've done a lot of work, can stronger conclusions about the strengths and weakness of each model be stated.

The global mean turnover time calculated from total soil carbon content and global respiration of 14 years predicted for YASSO is in line with the other CMIP-models, for which 5 out of 11 models had this global mean turnover rate less than 20 years (Todd-Brown et al., 2014). This is a metric that is generally used for the large scale models and therefore we also showed this number here. In the comparison to other CMIP models YAS doesn't seem to be that much off. But the reviewer is of course correct in his concern that the biased NPP is further contributing to this and we take this into account in our addition to the Conclusions (shown above).

To further have stronger conclusions about the comparison between the models, we have added the following text to the Discussion (lines 424-430):

"*Fig. S9 shows that the bias relative to FLUXCOM exists throughout most of the Northern Hemisphere and the tropics, but has only minor influence on the seasonal cycle of GPP. The high estimate of GPP will propagate into larger NPP, litter input and therefore also simulated heterotrophic respiration and soil carbon stocks. While this may contribute to a slightly larger simulated seasonal cycle of atmospheric $CO_2$ at northern stations, it is unlikely that this will affect our conclusions on the impact of the different soil formulations on the ability of JSBACH to simulate the seasonal cycle of heterotrophic respiration and the residence time of carbon in soil, and as a consequence, its ability to reproduce observed seasonal cycle of atmospheric $CO_2$ or its longterm trend.*"

and to the Conclusions (lines 532-539):

"*The drivers of YAS have larger variability in their values during the seasonal cycle, that causes a more pronounced seasonal cycle in the heterotrophic respiration with the current parameterization. Concerning the results this leads to unrealistic seasonal cycles of $CO_2$ in temperate regions and tropics and calls for model improvement. CBA showed less pronounced seasonal cycles of heterotrophic respiration, and had issues with $CO_2$ amplitude only in the northern high latitudes. The linear moisture dependence therefore seems justified, however it likely causes the Central Asian region to have too large carbon stocks. Whether this is caused by too high drought sensitivity or problems in the predicted soil moisture by JSBACH is difficult to judge. The too high amplitude in the northern high regions might be a result from the biases in the gross fluxes of the modeling system.*"

Minor and technical concerns:

Line 66, have the 'two soil models' been introduced outside of the abstract?

Thanks, we added this sentence before (line 66):

"*The JSBACH model has two distinct soil models implemented in it (CBALANCE and YASSO).*"

Line 228, how the models "show clear differences and similar behavior"? This statement is confusing.

Thank you for noticing this, we have corrected the sentence to the form it was meant to be in (line 226):

"*The global total magnitudes of Rh are comparable, while the seasonal cycles show clear differences, also visible in different latitudinal regions.*"

Line 240-250 & Line 395 are these regional sensitivities predictable based on the functions shown in Fig. S1? If so maybe these results can be contextualized based on the assumptions of environmental sensitivities illustrates in S1?

Thank you for this idea. We have added text to these parts by referring to the environmental sensitivities:

(lines 243-246)
"*In two of these regions with a negative relationship between alpha and R_h (located in high latitudes), the variability of alpha is quite small and the plot shows high scatter (Fig. S3). The shape of the $T_{soil}$ dependency on the CBA decomposition is exponential, and the relationship is significant, when the range of the $T_{soil}$ values is over 15 degrees, which is larger than what is occurring in the tropics (Fig. S4).*"

(lines 248-250)
"*In this region the correlation is still significant, but the variability of the precipitation is lower than in the other regions (Fig. S5). Therefore the exponential relationship (Fig. S1d, Eq. 5) between decomposition and precipitation does not  lead to a stronger linear relationship in this region.*"

(lines 251-253):
"*This region has only a small seasonal variation in air temperature and the values are also partly located in the temperature range, where the temperature sensitivity of decomposition is weaker (Fig. S6, Eq. 4).*"

Line 376 replace 'abundance' with 'concentration'

Replaced.

Fig 3. I like the consistency of using of the same colors for YAS and CBA results in many of the revised figures, but then using the same colors to different latitude bands in Fig 3 (S2 & S9) is confusing.

Thank you for pointing this out. We have changed the colors in the figures 3, S2 and S9, so that this would be now clearer.

Fig. 4. It's not obvious why the authors report turnover time anomalies that are subtracted from different baselines? Moreover, why not use the same color bar if the results are supposed to be normalized somehow? When characterizing the models, does just showing the raw turnover times (with a common colorbar) tell us more? I guess this kind of plot nicely shows soil moisture vs. temperature gradient in turnover time, is that the intent? Maybe flip the order of Figs 4 & 5 so the broader differences in the models are first highlighted?

We decided to use the anomalies, since they clearly show where the turnover times are highest and lowest. Therefore in this original figure the problematic high carbon stocks predicted by the CBA model in the mid-latitudes of the Northern Hemisphere were clearly visible in this figure (the disagreement to the observations is visible in the latitudinal gradient figure S12). We considered this to be the best way to display the spatial distribution of the turnover times of the two different models, as the absolute magnitude of the turnover times is now visible in Figure 5. Yes, indeed, the plot shows now clearly the decline of the turnover time with CBA in the dry regions of the mid-latitudes in the Northern Hemisphere and how the temperature is the limiting factor in cold regions for both of the models.

We flipped the order of figures 4 and 5, as suggested, and also added the same color bar to both of the models. We abstain from showing the raw turnover times on the map, as also suggested, since the current Fig. 4 now shows the absolute turnover times of the two models in different temperature regions and it is challenging to get the differences visible in the spatial patterns that we aimed to demonstrate here (of which we added few sentences more to the results, in lines 292-294):

"*The CBA model shows longer turnover times in Central Asia, where the moisture conditions limit the decomposition. However, the YAS model does not show so large anomalies in this region.*"

Fig. 5. I like this nod to the Koven et al. 2017 study, can the observationally derived turnover times (and their uncertainty) from that paper be included?

Sure, we estimated the turnover times from the fit in Fig. 2 in the Koven study for two temperatures and compare them to the model estimates from this study (lines 465-472):

*"The study by Koven et al. (2017)  provided an empirically based turnover time as a function of temperature. At 20 °C  this turnover time was approximately 11 ± 2 years, being closer to the estimate for the YAS model (calculated for values 19.5 - 20.5 °C, and their standard deviation), being 22 ± 21 years °C and much lower compared to the CBA estimate of 64 ± 37 years. In lower temperatures, at -15 °C, the empirically based turnover time is 200 ± 100 years, and YAS underestimates this with 82 ± 41 years (calculated for values -15.5 - (-14.5) °C), whereas the prediction by CBA is closer (150 ± 80 years). Therefore, the turnover times simulated with the YAS model are closer to the observations in warm temperatures, but the turnover times are too low in cold temperatures. CBA estimated too high turnover times in warm temperatures, but turnover times in colder temperatures were in the same order as the observations."*

Fig 6, 7 & associated text. I'm not really sure what the 'relative values of the seasonal cycle amplitudes' are illustrating or how they help inform the story being told here. Given their sparing definition and interpretation maybe these sub-panels should be removed?

The reason we chose to show 'relative seasonal amplitude' was because the deviations between the amplitudes were visible already from the seasonal cycle figure and we therefore wanted here to be able to visualize the trend they are showing. But, as pointed out by the reviewer, they are not contributing much to the main story line here, so we leave them out, as proposed.

When comparing results from the two soil models please use the same axes (e.g. left y-axis Fig S8).

Thank you, we've done this now.

[revised manuscript text omitted]

~~In addition to the seasonal cycle the temporal development of the seasonal cycle amplitude for the four sites is displayed in Fig. **??**b, d, f, h. We show this development for relative values of the seasonal cycle amplitudes to make the temporal development visible, since the values between the two different model formulations are so different. The correlation coefficients between observed and the different modelled time series are shown in Table S4. CTE better captured the interannual variation of the seasonal cycle amplitude than the CBA and YAS models, which perform comparably. The YAS model shows stronger interannual variation at Niwot Ridge (Fig. **??**d) and this is caused by the small magnitude of the seasonal cycle amplitude by YAS at this site.~~

[revised manuscript text omitted]